# Neural Thompson Sampling

**Weitong Zhang**
Department of Computer Science
University of California, Los Angeles
Los Angeles, CA, USA, 90095
wt.zhang@ucla.edu

**Dongruo Zhou**
Department of Computer Science
University of California, Los Angeles
Los Angeles, CA, USA, 90095
drzhou@cs.ucla.edu

**Lihong Li**
Google Research
USA
lihong@google.com

**Quanquan Gu**
Department of Computer Science
University of California, Los Angeles
Los Angeles, CA, USA, 90095
qgu@cs.ucla.edu

## Abstract

Thompson Sampling (TS) is one of the most effective algorithms for solving contextual multi-armed bandit problems. In this paper, we propose a new algorithm, called Neural Thompson Sampling, which adapts deep neural networks for both exploration and exploitation. At the core of our algorithm is a novel posterior distribution of the reward, where its mean is the neural network approximator, and its variance is built upon the neural tangent features of the corresponding neural network. We prove that, provided the underlying reward function is bounded, the proposed algorithm is guaranteed to achieve a cumulative regret of $\mathcal{O}(T^{1/2})$, which matches the regret of other contextual bandit algorithms in terms of total round number $T$. Experimental comparisons with other benchmark bandit algorithms on various data sets corroborate our theory.

## 1 Introduction

The stochastic multi-armed bandit (Bubeck & Cesa-Bianchi, 2012; Lattimore & Szepesvári, 2020) has been extensively studied, as an important model to optimize the trade-off between exploration and exploitation in sequential decision making. Among its many variants, the contextual bandit is widely used in real-world applications such as recommendation (Li et al., 2010), advertising (Graepel et al., 2010), robotic control (Mahler et al., 2016), and healthcare (Greenewald et al., 2017).

In each round of a contextual bandit, the agent observes a feature vector (the "context") for each of the $K$ arms, pulls one of them, and in return receives a scalar reward. The goal is to maximize the cumulative reward, or minimize the regret (to be defined later), in a total of $T$ rounds. To do so, the agent must find a trade-off between exploration and exploitation. One of the most effective and widely used techniques is *Thompson Sampling*, or TS (Thompson, 1933). The basic idea is to compute the posterior distribution of each arm being optimal for the present context, and sample an arm from this distribution. TS is often easy to implement, and has found great success in practice (Chapelle & Li, 2011; Graepel et al., 2010; Kawale et al., 2015; Russo et al., 2017).

Recently, a series of work has applied TS or its variants to explore in contextual bandits with neural network models (Blundell et al., 2015; Kveton et al., 2020; Lu & Van Roy, 2017; Riquelme et al., 2018). Riquelme et al. (2018) proposed NeuralLinear, which maintains a neural network and chooses the best arm in each round according to a Bayesian linear regression on top of the last network layer. Kveton et al. (2020) proposed DeepFPL, which trains a neural network based on perturbed training data and chooses the best arm in each round based on the neural network output. Similar approaches have also been used in more general reinforcement learning problem (e.g., Azizzadenesheli et al., 2018; Fortunato et al., 2018; Lipton et al., 2018; Osband et al., 2016a). Despite the reported empirical success, strong regret guarantees for TS remain limited to relatively simple models, under fairly restrictive assumptions on the reward function. Examples are linear

functions (Abeille & Lazaric, 2017; Agrawal & Goyal, 2013; Kocák et al., 2014; Russo & Van Roy, 2014), generalized linear functions (Kveton et al., 2020; Russo & Van Roy, 2014), or functions with small RKHS norm induced by a properly selected kernel (Chowdhury & Gopalan, 2017).

In this paper, we provide, to the best of our knowledge, the first near-optimal regret bound for neural network-based Thompson Sampling. Our contributions are threefold. First, we propose a new algorithm, *Neural Thompson Sampling (NeuralTS)*, to incorporate TS exploration with neural networks. It differs from NeuralLinear (Riquelme et al., 2018) by considering weight uncertainty in *all* layers, and from other neural network-based TS implementations (Blundell et al., 2015; Kveton et al., 2020) by sampling the estimated reward from the posterior (as opposed to sampling parameters).

Second, we give a regret analysis for the algorithm, and obtain an $\widetilde{\mathcal{O}}(\widetilde{d}\sqrt{T})$ regret, where $\widetilde{d}$ is the *effective dimension* and $T$ is the number of rounds. This result is comparable to previous bounds when specialized to the simpler, linear setting where the effective dimension coincides with the feature dimension (Agrawal & Goyal, 2013; Chowdhury & Gopalan, 2017).

Finally, we corroborate the analysis with an empirical evaluation of the algorithm on several benchmarks. Experiments show that NeuralTS yields competitive performance, in comparison with state-of-the-art baselines, thus suggest its practical value in addition to strong theoretical guarantees.

**Notation:**   Scalars and constants are denoted by lower and upper case letters, respectively. Vectors are denoted by lower case bold face letters $\mathbf{x}$, and matrices by upper case bold face letters $\mathbf{A}$. We denote by $[k]$ the set $\{1, 2, \cdots, k\}$ for positive integers $k$. For two non-negative sequence $\{a_n\}, \{b_n\}$, $a_n = \mathcal{O}(b_n)$ means that there exists a positive constant $C$ such that $a_n \leq Cb_n$, and we use $\widetilde{\mathcal{O}}(\cdot)$ to hide the log factor in $\mathcal{O}(\cdot)$. We denote by $\|\cdot\|_2$ the Euclidean norm of vectors and the spectral norm of matrices, and by $\|\cdot\|_{\mathrm{F}}$ the Frobenius norm of a matrix.

## 2   PROBLEM SETTING AND PROPOSED ALGORITHM

In this work, we consider contextual $K$-armed bandits, where the total number of rounds $T$ is known. At round $t \in [T]$, the agent observes $K$ contextual vectors $\{\mathbf{x}_{t,k} \in \mathbb{R}^d \mid k \in [K]\}$. Then the agent selects an arm $a_t$ and receives a reward $r_{t,a_t}$. Our goal is to minimize the following pseudo regret:

$$R_T = \mathbb{E}\bigg[\sum_{t=1}^{T}(r_{t,a_t^*} - r_{t,a_t})\bigg], \tag{2.1}$$

where $a_t^*$ is the optimal arm at round $t$ that has the maximum expected reward: $a_t^* = \mathrm{argmax}_{a\in[K]} \mathbb{E}[r_{t,a}]$. To estimate the unknown reward given a contextual vector $\mathbf{x}$, we use a fully connected neural network $f(\mathbf{x}; \boldsymbol{\theta})$ of depth $L \geq 2$, defined recursively by

$$\begin{aligned} f_1 &= \mathbf{W}_1\,\mathbf{x}, \\ f_l &= \mathbf{W}_l\,\mathrm{ReLU}(f_{l-1}), \quad 2 \leq l \leq L, \\ f(\mathbf{x}; \boldsymbol{\theta}) &= \sqrt{m}f_L, \end{aligned} \tag{2.2}$$

where $\mathrm{ReLU}(x) := \max\{x, 0\}$, $m$ is the width of neural network, $\mathbf{W}_1 \in \mathbb{R}^{m \times d}, \mathbf{W}_l \in \mathbb{R}^{m \times m}, 2 \leq l < L, \mathbf{W}_L \in \mathbb{R}^{1 \times m}$, $\boldsymbol{\theta} = (\mathrm{vec}(\mathbf{W}_1); \cdots; \mathrm{vec}(\mathbf{W}_L)) \in \mathbb{R}^p$ is the collection of parameters of the neural network, $p = dm + m^2(L-2) + m$, and $\mathbf{g}(\mathbf{x}; \boldsymbol{\theta}) = \nabla_{\boldsymbol{\theta}} f(\mathbf{x}; \boldsymbol{\theta})$ is the gradient of $f(\mathbf{x}; \boldsymbol{\theta})$ w.r.t. $\boldsymbol{\theta}$.

Our Neural Thompson Sampling is given in Algorithm 1. It maintains a Gaussian distribution for each arm's reward. When selecting an arm, it samples the reward of each arm from the reward's posterior distribution, and then pulls the greedy arm (lines 4–8). Once the reward is observed, it updates the posterior (lines 9 & 10). The mean of the posterior distribution is set to the output of the neural network, whose parameter is the solution to the following minimization problem:

$$\min_{\boldsymbol{\theta}} L(\boldsymbol{\theta}) = \sum_{i=1}^{t}[f(\mathbf{x}_{i,a_i}; \boldsymbol{\theta}) - r_{i,a_i}]^2/2 + m\lambda\|\boldsymbol{\theta} - \boldsymbol{\theta}_0\|_2^2/2. \tag{2.3}$$

We can see that (2.3) is an $\ell_2$-regularized square loss minimization problem, where the regularization term centers at the randomly initialized network parameter $\boldsymbol{\theta}_0$. We adapt gradient descent to solve (2.3) with step size $\eta$ and total number of iterations $J$.

---

**Algorithm 1** Neural Thompson Sampling (NeuralTS)

---

**Input:** Number of rounds $T$, exploration variance $\nu$, network width $m$, regularization parameter $\lambda$.
 1: Set $\mathbf{U}_0 = \lambda\mathbf{I}$
 2: Initialize $\boldsymbol{\theta}_0 = (\text{vec}(\mathbf{W}_1); \cdots ; \text{vec}(\mathbf{W}_L)) \in \mathbb{R}^p$, where for each $1 \leq l \leq L - 1$, $\mathbf{W}_l = (\mathbf{W}, \mathbf{0}; \mathbf{0}, \mathbf{W})$, each entry of $\mathbf{W}$ is generated independently from $N(0, 4/m)$; $\mathbf{W}_L = (\mathbf{w}^\top, -\mathbf{w}^\top)$, each entry of $\mathbf{w}$ is generated independently from $N(0, 2/m)$.
 3: **for** $t = 1, \cdots, T$ **do**
 4:  **for** $k = 1, \cdots, K$ **do**
 5:   $\sigma_{t,k}^2 = \lambda\, \mathbf{g}^\top(\mathbf{x}_{t,k}; \boldsymbol{\theta}_{t-1})\, \mathbf{U}_{t-1}^{-1}\, \mathbf{g}(\mathbf{x}_{t,k}; \boldsymbol{\theta}_{t-1})/m$
 6:   Sample estimated reward $\widetilde{r}_{t,k} \sim \mathcal{N}(f(\mathbf{x}_{t,k}; \boldsymbol{\theta}_{t-1}), \nu^2\sigma_{t,k}^2)$
 7:  **end for**
 8:  Pull arm $a_t$ and receive reward $r_{t,a_t}$, where $a_t = \text{argmax}_a\, \widetilde{r}_{t,a}$
 9:  Set $\boldsymbol{\theta}_t$ to be the output of gradient descent for solving (2.3)
10:  $\mathbf{U}_t = \mathbf{U}_{t-1} + \mathbf{g}(\mathbf{x}_{t,a_t}; \boldsymbol{\theta}_t)\mathbf{g}(\mathbf{x}_{t,a_t}; \boldsymbol{\theta}_t)^\top/m$
11: **end for**

---

A few observations about our algorithm are in place. First, compared to typical ways of implementing Thompson Sampling with neural networks, NeuralTS samples from the posterior distribution of the *scalar reward*, instead of the network parameters. It is therefore simpler and more efficient, as the number of parameters in practice can be large.

Second, the algorithm maintains the posterior distributions related to parameters of all layers of the network, as opposed to the last layer only (Riquelme et al., 2018). This difference is crucial in our regret analysis. It allows us to build a connection between Algorithm 1 and recent work about deep learning theory (Allen-Zhu et al., 2018; Cao & Gu, 2019), in order to obtain theoretical guarantees as will be shown in the next section.

Third, different from linear or kernelized TS (Agrawal & Goyal, 2013; Chowdhury & Gopalan, 2017), whose posterior can be computed in closed forms, NeuralTS solves a non-convex optimization problem (2.3) by gradient descent. This difference requires additional techniques in the regret analysis. Moreover, *stochastic* gradient descent can be used to solve the optimization problem with a similar theoretical guarantee (Allen-Zhu et al., 2018; Du et al., 2018; Zou et al., 2019). For simplicity of exposition, we will focus on the exact gradient descent approach.

## 3 REGRET ANALYSIS

In this section, we provide a regret analysis of NeuralTS. We assume that there exists an unknown reward function $h$ such that for any $1 \leq t \leq T$ and $1 \leq k \leq K$,

$$r_{t,k} = h(\mathbf{x}_{t,k}) + \xi_{t,k}, \quad \text{with} \quad |h(\mathbf{x}_{t,k})| \leq 1$$

where $\{\xi_{t,k}\}$ forms an $R$-sub-Gaussian martingale difference sequence with constant $R > 0$, i.e., $\mathbb{E}[\exp(\lambda\xi_{t,k})|\xi_{1:t-1,k}, \mathbf{x}_{1:t,k}] \leq \exp(\lambda^2 R^2)$ for all $\lambda \in \mathbb{R}$. Such an assumption on the noise sequence is widely adapted in contextual bandit literature (Agrawal & Goyal, 2013; Bubeck & Cesa-Bianchi, 2012; Chowdhury & Gopalan, 2017; Chu et al., 2011; Lattimore & Szepesvári, 2020; Valko et al., 2013).

Next, we provide necessary background on the neural tangent kernel (NTK) theory (Jacot et al., 2018), which plays a crucial role in our analysis. In the analysis, we denote by $\{\mathbf{x}^i\}_{i=1}^{TK}$ the set of observed contexts of all arms and all rounds: $\{\mathbf{x}_{t,k}\}_{1 \leq t \leq T, 1 \leq k \leq K}$ where $i = K(t - 1) + k$.

**Definition 3.1** (Jacot et al. (2018)). Define

$$\widetilde{\mathbf{H}}_{i,j}^{(1)} = \boldsymbol{\Sigma}_{i,j}^{(1)} = \langle\mathbf{x}^i, \mathbf{x}^j\rangle, \mathbf{A}_{i,j}^{(l)} = \begin{pmatrix} \boldsymbol{\Sigma}_{i,i}^{(l)} & \boldsymbol{\Sigma}_{i,j}^{(l)} \\ \boldsymbol{\Sigma}_{i,j}^{(l)} & \boldsymbol{\Sigma}_{j,j}^{(l)} \end{pmatrix},$$

$$\boldsymbol{\Sigma}_{i,j}^{(l+1)} = 2\mathbb{E}_{(u,v)\sim N(\mathbf{0}, \mathbf{A}_{i,j}^{(l)})}\max\{u, 0\}\max\{v, 0\},$$

$$\widetilde{\mathbf{H}}_{i,j}^{(l+1)} = 2\widetilde{\mathbf{H}}_{i,j}^{(l)}\mathbb{E}_{(u,v)\sim N(\mathbf{0}, \mathbf{A}_{i,j}^{(l)})}\mathbb{1}(u \geq 0)\mathbb{1}(v \geq 0) + \boldsymbol{\Sigma}_{i,j}^{(l+1)}.$$

Then, $\mathbf{H} = (\widetilde{\mathbf{H}}^{(L)} + \boldsymbol{\Sigma}^{(L)})/2$ is called the neural tangent kernel matrix on the context set.

The NTK technique builds a connection between deep neural networks and kernel methods. It enables us to adapt some complexity measures for kernel methods to describe the complexity of the neural network, as given by the following definition.

**Definition 3.2.** The *effective dimension* $\widetilde{d}$ of matrix $\mathbf{H}$ with regularization parameter $\lambda$ is defined as

$$\widetilde{d} = \frac{\log\det(\mathbf{I} + \mathbf{H}/\lambda)}{\log(1 + TK/\lambda)}.$$

**Remark 3.3.** The effective dimension is a metric to describe the actual underlying dimension in the set of observed contexts, and has been used by Valko et al. (2013) for the analysis of kernel UCB. Our definition here is adapted from Yang & Wang (2019), which also considers UCB-based exploration. Compared with the maximum information gain $\gamma_t$ used in Chowdhury & Gopalan (2017), one can verify that their Lemma 3 shows that $\gamma_t \geq \log\det(\mathbf{I} + \mathbf{H}/\lambda)/2$. Therefore, $\gamma_t$ and $\widetilde{d}$ are of the same order up to a ratio of $1/(2\log(1 + TK/\lambda))$. Furthermore, $\widetilde{d}$ can be upper bounded if all contexts $\mathbf{x}_i$ are nearly on some low-dimensional subspace of the RKHS space spanned by NTK (Appendix D).

We will make a regularity assumption on the contexts and the corresponding NTK matrix $\mathbf{H}$.

**Assumption 3.4.** Let $\mathbf{H}$ be defined in Definition 3.1. There exists $\lambda_0 > 0$, such that $\mathbf{H} \succeq \lambda_0 \mathbf{I}$. In addition, for any $t \in [T], k \in [K], \|\mathbf{x}_{t,k}\|_2 = 1$ and $[\mathbf{x}_{t,k}]_j = [\mathbf{x}_{t,k}]_{j+d/2}$.

The assumption that the NTK matrix is positive definite has been considered in prior work on NTK (Arora et al., 2019; Du et al., 2018). The assumption on context $\mathbf{x}_{t,a}$ ensures that the initial output of neural network $f(\mathbf{x}; \boldsymbol{\theta}_0)$ is 0 with the random initialization suggested in Algorithm 1. The condition on $\mathbf{x}$ is easy to satisfy, since for any context $\mathbf{x}$, one can always construct a new context $\widetilde{\mathbf{x}}$ as $[\mathbf{x}/(\sqrt{2}\|\mathbf{x}\|_2), \mathbf{x}/(\sqrt{2}\|\mathbf{x}\|_2)]^\top$.

We are now ready to present the main result of the paper:

**Theorem 3.5.** Under Assumption 3.4, set the parameters in Algorithm 1 as $\lambda = 1 + 1/T$, $\nu = B + R\sqrt{\widetilde{d}\log(1 + TK/\lambda) + 2 + 2\log(1/\delta)}$ where $B = \max\left\{1/(22\mathrm{e}\sqrt{\pi}), \sqrt{2\mathbf{h}^\top \mathbf{H}^{-1}\mathbf{h}}\right\}$ with $\mathbf{h} = (h(\mathbf{x}^1), \ldots, h(\mathbf{x}^{TK}))^\top$, and $R$ is the sub-Gaussian parameter. In line 9 of Algorithm 1, set $\eta = C_1(m\lambda + mLT)^{-1}$ and $J = (1 + LT/\lambda)(C_2 + \log(T^3 L\lambda^{-1}\log(1/\delta)))/C_1$ for some positive constants $C_1, C_2$. If the network width $m$ satisfies:

$$m \geq \mathrm{poly}\left(\lambda, T, K, L, \log(1/\delta), \lambda_0^{-1}\right),$$

then, with probability at least $1 - \delta$, the regret of Algorithm 1 is bounded as

$$R_T \leq C_3(1 + c_T)\nu\sqrt{2\lambda L(\widetilde{d}\log(1 + TK) + 1)T} + (4 + C_4(1 + c_T)\nu L)\sqrt{2\log(3/\delta)T} + 5,$$

where $C_3, C_4$ are some positive absolute constants, and $c_T = \sqrt{4\log T + 2\log K}$.

**Remark 3.6.** The definition $B$ in Theorem 3.5 is inspired by the RKHS norm of the reward function defined in Chowdhury & Gopalan (2017). It can be verified that when the reward function $h$ belongs to the function space induced by NTK, i.e., $\|h\|_{\mathcal{H}} < \infty$, we have $\sqrt{\mathbf{h}^\top \mathbf{H}^{-1}\mathbf{h}} \leq \|h\|_{\mathcal{H}}$ according to Zhou et al. (2019), which suggests that $B \leq \max\{1/(22\mathrm{e}\sqrt{\pi}), \sqrt{2}\|h\|_{\mathcal{H}}\}$.

**Remark 3.7.** Theorem 3.5 implies the regret of NeuralTS is on the order of $\widetilde{O}(\widetilde{d}T^{1/2})$. This result matches the state-of-the-art regret bound in Chowdhury & Gopalan (2017); Agrawal & Goyal (2013); Zhou et al. (2019); Kveton et al. (2020).

**Remark 3.8.** In Theorem 3.5, the requirement of $m$ is specified in Condition 4.1 and the proof of Theorem 3.5, which is a high-degree polynomial in the time horizon $T$, number of layers $L$ and number of actions $K$. However, in our experiments, we can choose reasonably small $m$ (e.g., $m = 100$) to obtain good performance of NeuralTS. See Appendix A.1 for more details. This discrepancy between theory and practice is due to the limitation of current NTK theory (Du et al., 2018; Allen-Zhu et al., 2018; Zou et al., 2019). Closing the gap is a venue for future work.

**Remark 3.9.** Theorem 3.5 suggests that we need to know $T$ before we run the algorithm in order to set $m$. When $T$ is unknown, we can use the standard doubling trick (See e.g., Cesa-Bianchi & Lugosi (2006)) to set $m$ adaptively. In detail, we decompose the time interval $(0, +\infty)$ as a union of non-overlapping intervals $[2^s, 2^{s+1})$. When $2^s \leq t < 2^{s+1}$, we restart NeuralTS with the input $T = 2^{s+1}$. It can be verified that similar $\widetilde{O}(\widetilde{d}\sqrt{T})$ regret still holds.

## 4 PROOF OF THE MAIN THEOREM

This section sketches the proof of Theorem 3.5, with supporting lemmas and technical details provided in Appendix B. While the proof roadmap is similar to previous work on Thompson Sampling (e.g., Agrawal & Goyal, 2013; Chowdhury & Gopalan, 2017; Kocák et al., 2014; Kveton et al., 2020), our proof needs to carefully track the approximation error of neural networks for approximating the reward function. To control the approximation error, the following condition on the neural network width is required in several technical lemmas.

**Condition 4.1.** The network width $m$ satisfies

$$m \geq C \max \left\{ \sqrt{\lambda} L^{-3/2} [\log(TKL^2/\delta)]^{3/2}, T^6 K^6 L^6 \log(TKL/\delta) \max\{\lambda_0^{-4}, 1\}, \right\}$$

$$m[\log m]^{-3} \geq CTL^{12}\lambda^{-1} + CT^7\lambda^{-8}L^{18}(\lambda + LT)^6 + CL^{21}T^7\lambda^{-7}(1 + \sqrt{T/\lambda})^6,$$

where $C$ is a positive absolute constant.

For any $t$, we define an event $\mathcal{E}_t^\sigma$ as follows

$$\mathcal{E}_t^\sigma = \{\omega \in \mathcal{F}_{t+1} : \forall k \in [K], \quad |\widetilde{r}_{t,k} - f(\mathbf{x}_{t,k}; \boldsymbol{\theta}_{t-1})| \leq c_t \nu \sigma_{t,k}\}, \tag{4.1}$$

where $c_t = \sqrt{4\log t + 2\log K}$. Under event $\mathcal{E}_t^\sigma$, the difference between the sampled reward $\widetilde{r}_{t,k}$ and the estimated mean reward $f(\mathbf{x}_{t,k}; \boldsymbol{\theta}_{t-1})$ can be controlled by the reward's posterior variance.

We also define an event $\mathcal{E}_t^\mu$ as follows

$$\mathcal{E}_t^\mu = \{\omega \in \mathcal{F}_t : \forall k \in [K], \quad |f(\mathbf{x}_{t,k}; \boldsymbol{\theta}_{t-1}) - h(\mathbf{x}_{t,k})| \leq \nu \sigma_{t,k} + \epsilon(m)\}, \tag{4.2}$$

where $\epsilon(m)$ is defined as

$$\epsilon(m) = \epsilon_p(m) + C_{\epsilon,1}(1 - \eta m\lambda)^J \sqrt{TL/\lambda}$$
$$\epsilon_p(m) = C_{\epsilon,2}T^{2/3}m^{-1/6}\lambda^{-2/3}L^3\sqrt{\log m} + C_{\epsilon,3}m^{-1/6}\sqrt{\log m}L^4T^{5/3}\lambda^{-5/3}(1 + \sqrt{T/\lambda})$$
$$+ C_{\epsilon,4}\Big(B + R\sqrt{\log \det(\mathbf{I} + \mathbf{H}/\lambda) + 2 + 2\log(1/\delta)}\Big)\sqrt{\log m}T^{7/6}m^{-1/6}\lambda^{-2/3}L^{9/2}, \tag{4.3}$$

and $\{C_{\epsilon,i}\}_{i=1}^4$ are some positive absolute constants. Under event $\mathcal{E}_t^\mu$, the estimated mean reward $f(\mathbf{x}_{t,k}; \boldsymbol{\theta}_{t-1})$ based on the neural network is similar to the true expected reward $h(\mathbf{x}_{t,k})$. Note that the additional term $\epsilon(m)$ is the approximate error of the neural networks for approximating the true reward function. This is a key difference in our proof from previous regret analysis of Thompson Sampling Agrawal & Goyal (2013); Chowdhury & Gopalan (2017), where there is no approximation error.

The following two lemmas show that both events $\mathcal{E}_t^\sigma$ and $\mathcal{E}_t^\mu$ happen with high probability.

**Lemma 4.2.** For any $t \in [T]$, $\Pr\left(\mathcal{E}_t^\sigma \big| \mathcal{F}_t\right) \geq 1 - t^{-2}$.

**Lemma 4.3.** Suppose the width of the neural network $m$ satisfies Condition 4.1. Set $\eta = C(m\lambda + mLT)^{-1}$, then we have $\Pr\left(\forall t \in [T], \mathcal{E}_t^\mu\right) \geq 1 - \delta$, where $C$ is an positive absolute constant.

The next lemma gives a lower bound of the probability that the sampled reward $\widetilde{r}$ is larger than true reward up to the approximation error $\epsilon(m)$.

**Lemma 4.4.** For any $t \in [T]$, $k \in [K]$, we have $\Pr\left(\widetilde{r}_{t,k} + \epsilon(m) > h(\mathbf{x}_{t,k}) \big| \mathcal{F}_t, \mathcal{E}_t^\mu\right) \geq (4e\sqrt{\pi})^{-1}$.

Following Agrawal & Goyal (2013), for any time $t$, we divide the arms into two groups: saturated and unsaturated arms, based on whether the standard deviation of the estimates for an arm is smaller than the standard deviation for the optimal arm or not. Note that the optimal arm is included in the group of unsaturated arms. More specifically, we define the set of saturated arms $S_t$ as follows

$$S_t = \big\{k \big| k \in [K], h(\mathbf{x}_{t,a_t^*}) - h(\mathbf{x}_{t,k}) \geq (1 + c_t)\nu\sigma_{t,k} + 2\epsilon(m)\big\}. \tag{4.4}$$

Note that we have taken the approximate error $\epsilon(m)$ into consideration when defining saturated arms, which differs from the Thompson Sampling literature (Agrawal & Goyal, 2013; Chowdhury & Gopalan, 2017). It is now easy to show that the immediate regret of playing an unsaturated arm can be bounded by the standard deviation plus the approximation error $\epsilon(m)$.

The following lemma shows that the probability of pulling a saturated arm is small in Algorithm 1.

**Lemma 4.5.** Let $a_t$ be the arm pulled at round $t \in [T]$. Then, $\Pr\left(a_t \notin S_t | \mathcal{F}_t, \mathcal{E}_t^\mu\right) \geq \frac{1}{4e\sqrt{\pi}} - \frac{1}{t^2}$.

The next lemma bounds the expectation of the regret at each round conditioned on $\mathcal{E}_t^\mu$.

**Lemma 4.6.** Suppose the width of the neural network $m$ satisfies Condition 4.1. Set $\eta = C_1(m\lambda + mLT)^{-1}$, then with probability at least $1 - \delta$, we have for all $t \in [T]$ that
$$\mathbb{E}[h(\mathbf{x}_{t,a_t^*}) - h(\mathbf{x}_{t,a_t})|\mathcal{F}_t, \mathcal{E}_t^\mu] \leq C_2(1 + c_t)\nu\sqrt{L}\mathbb{E}[\min\{\sigma_{t,a_t}, 1\}|\mathcal{F}_t, \mathcal{E}_t^\mu] + 4\epsilon(m) + 2t^{-2},$$
where $C_1, C_2$ are some positive absolute constants.

Based on Lemma 4.6, we define $\bar{\Delta}_t := (h(\mathbf{x}_{t,a_t^*}) - h(\mathbf{x}_{t,a_t}))\mathbb{1}(\mathcal{E}_t^\mu)$, and

$$X_t := \bar{\Delta}_t - \left(C_\Delta(1 + c_t)\nu\sqrt{L}\min\{\sigma_{t,a_t}, 1\} + 4\epsilon(m) + 2t^{-2}\right), \quad Y_t = \sum_{i=1}^t X_i, \tag{4.5}$$

where $C_\Delta$ is the same with constant $C$ in Lemma 4.6. By Lemma 4.6, we can verify that with probability at least $1 - \delta$, $\{Y_t\}$ forms a super martingale sequence since $\mathbb{E}(Y_t - Y_{t-1}) = \mathbb{E}X_t \leq 0$. By Azuma-Hoeffding inequality (Hoeffding, 1963), we can prove the following lemma.

**Lemma 4.7.** Suppose the width of the neural network $m$ satisfies Condition 4.1. Then set $\eta = C_1(m\lambda + mLT)^{-1}$, we have, with probability at least $1 - \delta$, that

$$\sum_{i=1}^T \bar{\Delta}_i \leq 4T\epsilon(m) + \pi^2/3 + C_2(1 + c_T)\nu\sqrt{L}\sum_{i=1}^T \min\{\sigma_{t,a_t}, 1\}$$
$$+ (4 + C_3(1 + c_T)\nu L + 4\epsilon(m))\sqrt{2\log(1/\delta)T},$$

where $C_1, C_2, C_3$ are some positive absolute constants.

The last lemma is used to control $\sum_{i=1}^T \min\{\sigma_{t,a_t}, 1\}$ in Lemma 4.7.

**Lemma 4.8.** Suppose the width of the neural network $m$ satisfies Condition 4.1. Then set $\eta = C_1(m\lambda + mLT)^{-1}$, we have, with probability at least $1 - \delta$, it holds that

$$\sum_{i=1}^T \min\{\sigma_{t,a_t}, 1\} \leq \sqrt{2\lambda T(\widetilde{d}\log(1 + TK) + 1)} + C_2 T^{13/6}\sqrt{\log m}\, m^{-1/6}\lambda^{-2/3}L^{9/2},$$

where $C_1, C_2$ are some positive absolute constants.

With all the above lemmas, we are ready to prove Theorem 3.5.

*Proof of Theorem 3.5.* By Lemma 4.3, $\mathcal{E}_t^\mu$ holds for all $t \in [T]$ with probability at least $1 - \delta$. Therefore, with probability at least $1 - \delta$, we have

$$R_T = \sum_{i=1}^T (h(\mathbf{x}_{t,a_t^*}) - h(\mathbf{x}_{t,a_t}))\mathbb{1}(\mathcal{E}_t^\mu)$$

$$\leq 4T\epsilon(m) + \frac{\pi^2}{3} + \bar{C}_1(1 + c_T)\nu\sqrt{L}\sum_{i=1}^T \min\{\sigma_{t,a_t}, 1\}$$

$$+ (4 + \bar{C}_2(1 + c_T)\nu L + 4\epsilon(m))\sqrt{2\log(1/\delta)T}$$

$$\leq \bar{C}_1(1 + c_T)\nu\sqrt{L}\left(\sqrt{2\lambda T(\widetilde{d}\log(1 + TK) + 1)} + \bar{C}_3 T^{13/6}\sqrt{\log m}\, m^{-1/6}\lambda^{-2/3}L^{9/2}\right)$$

$$+ \frac{\pi^2}{3} + 4T\epsilon(m) + 4\epsilon(m)\sqrt{2\log(1/\delta)T} + (4 + \bar{C}_2(1 + c_T)\nu L)\sqrt{2\log(1/\delta)T},$$

$$= \bar{C}_1(1 + c_T)\nu\sqrt{L}\left(\sqrt{2\lambda T(\widetilde{d}\log(1 + TK) + 1)} + \bar{C}_3 T^{13/6}\sqrt{\log m}\, m^{-1/6}\lambda^{-2/3}L^{9/2}\right)$$

$$+ \frac{\pi^2}{3} + \epsilon_p(m)(4T + \sqrt{2\log(1/\delta)T}) + (4 + \bar{C}_2(1 + c_T)\nu L)\sqrt{2\log(1/\delta)T}$$

$$+ C_{\epsilon,1}(1 - \eta m\lambda)^J\sqrt{TL/\lambda}(4T + \sqrt{2\log(1/\delta)T}),$$

where $\bar{C}_1, \bar{C}_2, \bar{C}_3$ are some positive absolute constants, the first inequality is due to Lemma 4.7, and the second inequality is due to Lemma 4.8. The third equation is from (4.3). By setting $\eta = \bar{C}_4(m\lambda + mLT)^{-1}$ and $J = (1 + LT/\lambda)(\log(24C_{\epsilon,1}) + \log(T^3L\lambda^{-1}\log(1/\delta)))/\bar{C}_4$, we have

$$C_{\epsilon,1}(1 - \eta m\lambda)^J \sqrt{TL/\lambda}(4T + \sqrt{2\log(1/\delta)T}) \leq \frac{1}{3},$$

Then choosing $m$ such that

$$\bar{C}_1\bar{C}_3(1 + c_T)\nu T^{13/6}\sqrt{\log m} m^{-1/6}\lambda^{-2/3}L^5 \leq \frac{1}{3}, \quad \epsilon_p(m)(4T + \sqrt{2\log(1/\delta)T}) \leq \frac{1}{3}.$$

$R_T$ can be further bounded by

$$R_T \leq \bar{C}_1(1 + c_T)\nu\sqrt{2\lambda L(\widetilde{d}\log(1 + TK) + 1)T} + (4 + \bar{C}_2(1 + c_T)\nu L)\sqrt{2\log(1/\delta)T} + 5.$$

Taking union bound over Lemmas 4.3, 4.7 and 4.8, the above inequality holds with probability $1 - 3\delta$. By replacing $\delta$ with $\delta/3$ and rearranging terms, we complete the proof. □

## 5 EXPERIMENTS

This section gives an empirical evaluation of our algorithm in several public benchmark datasets, including `adult`, `covertype`, `magic telescope`, `mushroom` and `shuttle`, all from UCI (Dua & Graff, 2017), as well as `MNIST` (LeCun et al., 2010). The algorithm is compared to several typical baselines: linear and kernelized Thompson Sampling (Agrawal & Goyal, 2013; Chowdhury & Gopalan, 2017), linear and kernelized UCB (Chu et al., 2011; Valko et al., 2013), BootstrapNN (Osband et al., 2016b; Riquelme et al., 2018), and $\epsilon$-greedy for neural networks. BootstrapNN trains multiple neural networks with subsampled data, and at each step pulls the greedy action based on a randomly selected network. It has been proposed as a way to approximate Thompson Sampling (Osband & Van Roy, 2015; Osband et al., 2016b).

### 5.1 EXPERIMENT SETUP

To transform these classification problems into multi-armed bandits, we adapt the disjoint models (Li et al., 2010) to build a context feature vector for each arm: given an input feature $\mathbf{x} \in \mathbb{R}^d$ of a $k$-class classification problem, we build the context feature vector with dimension $kd$ as: $\mathbf{x}_1 = (\mathbf{x}; \mathbf{0}; \cdots; \mathbf{0}), \mathbf{x}_2 = (\mathbf{0}; \mathbf{x}; \cdots; \mathbf{0}), \cdots, \mathbf{x}_k = (\mathbf{0}; \mathbf{0}; \cdots; \mathbf{x})$. Then, the algorithm generates a set of predicted reward following Algorithm 1 and pulls the greedy arm. For these classification problems, if the algorithm selects a correct class by pulling the corresponding arm, it will receive a reward as 1, otherwise 0. The cumulative regret over time horizon $T$ is measured by the total mistakes made by the algorithm. All experiments are repeated 8 times with reshuffled data.

We set the time horizon of our algorithm to $10\,000$ for all data sets, except for `mushroom` which contains only $8\,124$ data. In order to speed up training for the NeuralUCB and Neural Thompson Sampling, we use the inverse of the diagonal elements of $\mathbf{U}$ as an approximation of $\mathbf{U}^{-1}$. Also, since calculating the kernel matrix is expensive, we stop training at $t = 1000$ and keep evaluating the performance for the rest of the time, similar to previous work (Riquelme et al., 2018; Zhou et al., 2019). Due to space limit, we defer the results on `adult`, `covertype` and `magic telescope`, as well as further experiment details, to Appendix A. In this section, we only show the results on `mushroom`, `shuttle` and `MNIST`.

### 5.2 EXPERIMENT I: PERFORMANCE OF NEURAL THOMPSON SAMPLING

The experiment results of Neural Thompson Sampling and other benchmark algorithms are shown in Figure 1. A few observations are in place. First, Neural Thompson Sampling's performance is among the best in 6 datasets and is significantly better than all other baselines in 2 of them. Second, the function class used by an algorithm is important. Those with linear representations tend to perform worse due to the nonlinearity of rewards in the data. Third, Thompson Sampling is competitive with, and sometimes better than, other exploration strategies with the same function class, in particular when neural networks are used.

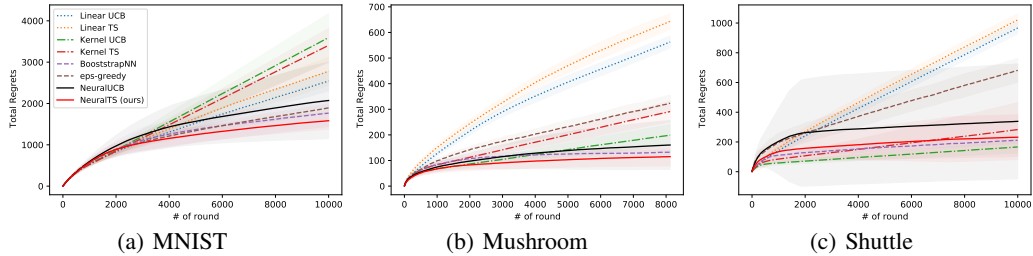

Figure 1: Comparison of Neural Thompson Sampling and baselines on UCI datasets and MNIST dataset. The total regret measures cumulative classification errors made by an algorithm. Results are averaged over 8 runs with standard errors shown as shaded areas.

## 5.3 EXPERIMENT II: ROBUSTNESS TO REWARD DELAY

This experiment is inspired by practical scenarios where reward signals are delayed, due to various constraints, as described by Chapelle & Li (2011). We study how robust the two most competitive methods from Experiment I, Neural UCB and Neural Thompson Sampling, are when rewards are delayed. More specifically, the reward after taking an action is not revealed immediately, but arrive in batches when the algorithms will update their models. The experiment setup is otherwise identical to Experiment I. Here, we vary the batch size (i.e., the amount of reward delay), and Figure 2 shows the corresponding total regret. Clearly, we recover the result in Experiment I when the delay is 0. Consistent with previous findings (Chapelle & Li, 2011), Neural TS degrades much more gracefully than Neural UCB when the reward delay increases. The benefit may be explained by the algorithm's randomized exploration nature that encourages exploration between batches. We, therefore, expect wider applicability of Neural TS in practical applications.

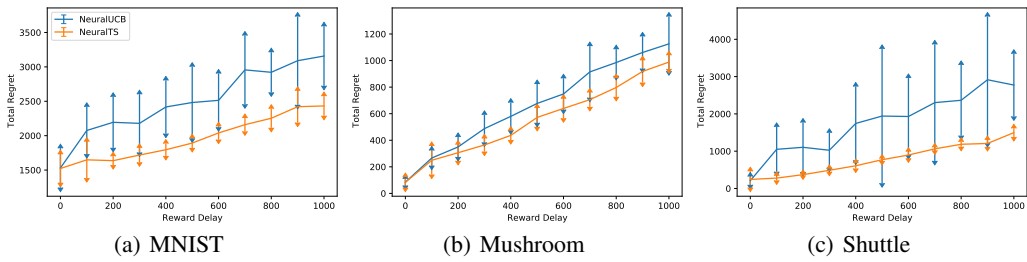

Figure 2: Comparison of Neural Thompson Sampling and Neural UCB on UCI datasets and MNIST dataset under different scale of delay. The total regret measures cumulative classification errors made by an algorithm. Results are averaged over 8 runs with standard errors shown as error bar.

## 6 RELATED WORK

Thompson Sampling was proposed as an exploration heuristic almost nine decades ago (Thompson, 1933), and has received significant interest in the last decade. Previous works related to the present paper are discussed in the introduction, and are not repeated here.

Upper confidence bound or UCB (Agrawal, 1995; Auer et al., 2002; Lai & Robbins, 1985) is a widely used alternative to Thompson Sampling for exploration. This strategy is shown to achieve near-optimal regrets in a range of settings, such as linear bandits (Abbasi-Yadkori et al., 2011; Auer, 2002; Chu et al., 2011), generalized linear bandits (Filippi et al., 2010; Jun et al., 2017; Li et al., 2017), and kernelized contextual bandits (Valko et al., 2013).

Neural networks are increasingly used in contextual bandits. In addition to those mentioned earlier (Blundell et al., 2015; Kveton et al., 2020; Lu & Van Roy, 2017; Riquelme et al., 2018), Zahavy

& Mannor (2019) used a deep neural network to provide a feature mapping and explored only at the last layer. Schwenk & Bengio (2000) proposed an algorithm by boosting the estimation of multiple deep neural networks. While these methods all show promise empirically, no regret guarantees are known. Recently, Foster & Rakhlin (2020) proposed a special regression oracle and randomized exploration for contextual bandits with a general function class (including neural networks) along with theoretical analysis. Zhou et al. (2019) proposed a neural UCB algorithm with near-optimal regret based on UCB exploration, while this paper focuses on Thompson Sampling.

## 7    CONCLUSIONS

In this paper, we adapt Thompson Sampling to neural networks. Building on recent advances in deep learning theory, we are able to show that the proposed algorithm, NeuralTS, enjoys a $\widetilde{\mathcal{O}}(\widetilde{d}T^{1/2})$ regret bound. We also show the algorithm works well empirically on benchmark problems, in comparison with multiple strong baselines.

The promising results suggest a few interesting directions for future research. First, our analysis needs NeuralTS to perform multiple gradient descent steps to train the neural network in each round. It is interesting to analyze the case where NeuralTS only performs one gradient descent step in each round, and in particular, the trade-off between optimization precision and regret minimization. Second, when the number of arms is finite, $\widetilde{\mathcal{O}}(\sqrt{dT})$ regret has been established for parametric bandits with linear and generalized linear reward functions. It is an open problem how to adapt NeuralTS to achieve the same rate. Third, Allen-Zhu & Li (2019) suggested that neural networks may behave differently from a neural tangent kernel under some parameter regimes. It is interesting to investigate whether similar results hold for neural contextual bandit algorithms like NeuralTS.

### ACKNOWLEDGEMENT

We would like to thank the anonymous reviewers for their helpful comments. WZ, DZ and QG are partially supported by the National Science Foundation CAREER Award 1906169 and IIS-1904183. The views and conclusions contained in this paper are those of the authors and should not be interpreted as representing any funding agencies.

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

# A   FURTHER DETAIL OF THE EXPERIMENTS IN SECTION 5

## A.1   PARAMETER TUNING

In the experiments, we shuffle all datasets randomly, and normalize the features so that their $\ell_2$-norm is unity. One-hidden-layer neural networks with 100 neurons are used. Note that we do not choose $m$ as suggested by theory, and such a disconnection has its root in the current deep learning theory based on neural tangent kernel, which is not specific in this work. During posterior updating, gradient descent is run for 100 iterations with learning rate 0.001. For BootstrapNN, we use 10 identical networks, and to train each network, data point at each round has probability 0.8 to be included for training ($p = 10, q = 0.8$ in the original paper (Schwenk & Bengio, 2000)) For $\epsilon$-Greedy, we tune $\epsilon$ with a grid search on $\{0.01, 0.05, 0.1\}$. For $(\lambda, \nu)$ used in linear and kernel UCB / Thompson Sampling, we set $\lambda = 1$ following previous works (Agrawal & Goyal, 2013; Chowdhury & Gopalan, 2017), and do a grid search of $\nu \in \{1, 0.1, 0.01\}$ to select the parameter with best performance. For the Neural UCB / Thompson Sampling methods, we use a grid search on $\lambda \in \{1, 10^{-1}, 10^{-2}, 10^{-3}\}$ and $\nu \in \{10^{-1}, 10^{-2}, 10^{-3}, 10^{-4}, 10^{-5}\}$. All experiments are repeated 20 times, and the average and standard error are reported.

## A.2   DETAILED RESULTS

Table 1 summarizes the total regrets measured at the last round on different data sets, with mean and standard deviation error computed based on 20 independent runs. The **Bold Faced** data is the top performance over 8 experiments. Table 2 shows the number of times the algorithm in that row significantly outperforms, ties, or significantly underperforms, compared with other algorithm with $t$-test at 90% significance level. Figure 3 shows the performance of Neural Thompson Sampling compared with other baseline method. Figure 4 shows the comparison between Neural Thompson Sampling and Neural UCB in delay reward settings.

Table 1: Total regrets get at the last step with standard deviation attached

|  | Adult | Covertype | Magic[1] | MNIST | Mushroom | Shuttle |
|---|---|---|---|---|---|---|
| Round# | 10 000 | 10 000 | 10 000 | 10 000 | 8 124 | 10 000 |
| Input Dim[2] | $2 \times 15$ | $2 \times 55$ | $2 \times 12$ | $10 \times 784$ | $2 \times 23$ | $7 \times 9$ |
| Random[3] | 5000 | 5000 | 5000 | 9000 | 4062 | 8571 |
| Linear UCB | 2097.5 $\pm 50.3$ | 3222.7 $\pm 67.2$ | 2604.4 $\pm 34.6$ | 2544.0 $\pm 235.4$ | 562.7 $\pm 23.1$ | 966.6 $\pm 39.0$ |
| Linear TS | 2154.7 $\pm 40.5$ | 4297.3 $\pm 328.7$ | 2700.5 $\pm 46.7$ | 2781.4 $\pm 338.3$ | 643.3 $\pm 30.4$ | 1020.9 $\pm 42.8$ |
| Kernel UCB | 2080.1 $\pm 44.8$ | 3546.2 $\pm 175.7$ | 2406.5 $\pm 79.4$ | 3595.8 $\pm 580.1$ | 199.0 $\pm 41.0$ | **166.5** $\pm\mathbf{39.4}$ |
| Kernel TS | 2111.5 $\pm 87.4$ | 3659.9 $\pm 113.8$ | 2442.6 $\pm 64.5$ | 3406.0 $\pm 411.7$ | 291.2 $\pm 40.0$ | 283.3 $\pm 180.5$ |
| BooststrapNN | 2097.3 $\pm 39.3$ | 3067.0 $\pm 56.1$ | 2269.4 $\pm 27.9$ | 1765.6 $\pm 321.1$ | 132.3 $\pm 8.6$ | 211.7 $\pm 20.9$ |
| eps-greedy | 2328.5 $\pm 50.4$ | 3334.2 $\pm 72.6$ | 2381.8 $\pm 37.3$ | 1893.2 $\pm 93.7$ | 323.2 $\pm 32.5$ | 682.0 $\pm 79.8$ |
| NeuralUCB | 2061.8 $\pm 42.8$ | 3012.1 $\pm 87.0$ | **2033.0** $\pm\mathbf{48.6}$ | 2071.6 $\pm 922.2$ | 160.4 $\pm 95.3$ | 338.6 $\pm 386.4$ |
| NeuralTS (ours) | **2092.5** $\pm\mathbf{48.0}$ | **2999.1** $\pm\mathbf{74.3}$ | 2037.4 $\pm 61.3$ | **1583.4** $\pm\mathbf{198.5}$ | **115.0** $\pm\mathbf{35.8}$ | 232.0 $\pm 149.5$ |

---

[1] Magic is short for data set MagicTelescope

[2] Using disjoint encoding thus is `NumofClass × NumofFeatures`

[3] Random pulling an arm at each round

[4] Magic is short for data set MagicTelescope

Table 2: Performance on total regret comparing with other methods on all datasets. Tuple $(w/t/l)$ indicates the times of the algorithm at that row $w$ins, $t$ies with or $l$oses, compared to all other 7 algorithms with $t$-test at 90% significant level.

|  | Adult | Covertype | Magic[4] | MNIST | Mushroom | Shuttle |
|---|---|---|---|---|---|---|
| Linear UCB | 2/3/2 | 4/0/3 | 1/0/6 | 2/2/3 | 1/0/6 | 1/0/6 |
| Linear TS | 1/0/6 | 0/0/7 | 0/0/7 | 2/1/4 | 0/0/7 | 0/0/7 |
| Kernel UCB | 4/3/0 | 2/0/5 | 3/1/3 | 0/0/7 | 4/0/3 | 7/0/0 |
| Kernel TS | 2/3/2 | 1/0/6 | 2/0/5 | 1/0/6 | 3/0/4 | 3/3/1 |
| BooststrapNN | 2/4/1 | 5/0/2 | 5/0/2 | 4/3/0 | 5/2/0 | 3/3/1 |
| eps-greedy | 0/0/7 | 3/0/4 | 3/1/3 | 4/2/1 | 2/0/5 | 2/0/5 |
| NeuralUCB | 6/1/0 | 6/1/0 | 6/1/0 | 3/3/1 | 5/1/1 | 3/3/1 |
| NeuralTS (ours) | 2/4/1 | 6/1/0 | 6/1/0 | 6/1/0 | 6/1/0 | 3/3/1 |

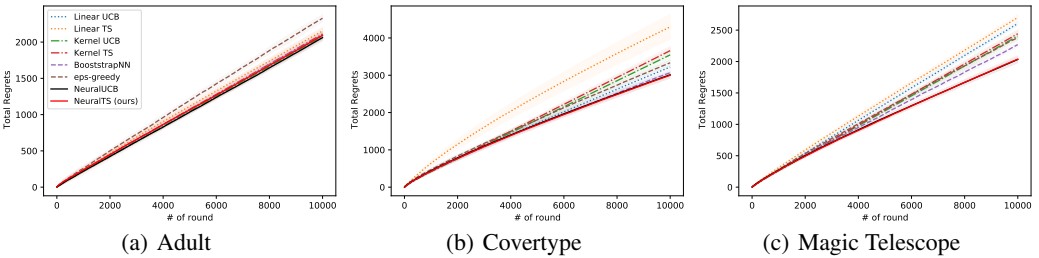

(a) Adult      (b) Covertype      (c) Magic Telescope

Figure 3: Comparison of Neural Thompson Sampling and baselines on UCI datasets and MNIST dataset. The total regret measures cumulative classification errors made by an algorithm. Results are averaged over multiple runs with standard errors shown as shaded areas.

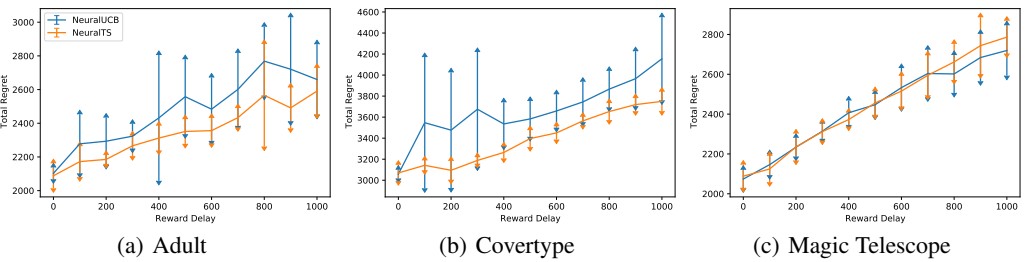

(a) Adult      (b) Covertype      (c) Magic Telescope

Figure 4: Comparison of Neural Thompson Sampling and Neural UCB on UCI datasets and MNIST dataset under different scale of delay. The total regret measures cumulative classification errors made by an algorithm. Results are averaged over multiple runs with standard errors shown as error bar.

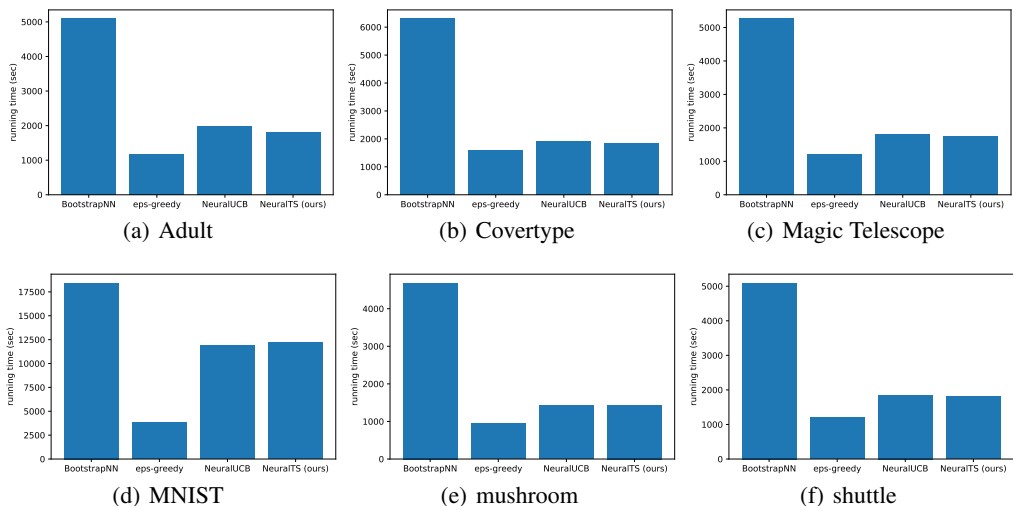

Figure 5: Comparison of the running time for Neural TS, Neural UCB and $\epsilon$-greedy for neural networks on UCI datasets and MNIST dataset.

### A.3 RUN TIME ANALYSIS

We compare the run time of the four algorithms based on neural networks: BootstrapNN, $\epsilon$-greedy for neural networks, NeuralUCB, and NeuralTS. The comparison is shown in Figure 5. We can see that NeuralTS and NeuralUCB are about 2 to 3 times slower than $\epsilon$-greedy, which is due to the extra calculation of the neural network gradient for each input context. BootstrapNN is often more than 5 times slower than $\epsilon$-greedy because it has to train several neural networks at each round.

## B PROOF OF LEMMAS IN SECTION 4

Under Condition 4.1, we can show that the following inequalities hold.

$$2\sqrt{1/\lambda} \geq C_{m,1} m^{-1} L^{-3/2} [\log(TKL^2/\delta)]^{3/2},$$
$$2\sqrt{T/\lambda} \leq C_{m,2} \min \left\{ m^{1/2} L^{-6} [\log m]^{-3/2}, m^{7/8} \left( (\lambda\eta)^2 L^{-6} T^{-1} (\log m)^{-1} \right)^{3/8} \right\},$$
$$m^{1/6} \geq C_{m,3} \sqrt{\log m} L^{7/2} T^{7/6} \lambda^{-7/6} (1 + \sqrt{T/\lambda})$$
$$m \geq C_{m,4} T^6 K^6 L^6 \log(TKL/\delta) \max\{\lambda_0^{-4}, 1\},$$

where $\{C_{m,1}, C_{m,2}, \ldots, C_{m,4}\}$ are some positive absolute constants.

### B.1 PROOF OF LEMMA 4.2

The following concentration bound on Gaussian distributions will be useful in our proof.

**Lemma B.1** (Hoffman et al. (2013)). Consider a normally distributed random variable $X \sim \mathcal{N}(\mu, \sigma^2)$ and $\beta \geq 0$. The probability that $X$ is within a radius of $\beta\sigma$ from its mean can then be written as

$$\Pr \left( |X - \mu| \leq \beta\sigma \right) \geq 1 - \exp(-\beta^2/2).$$

*Proof of Lemma 4.2.* Since the estimated reward $\widetilde{r}_{t,k}$ is sampled from $\mathcal{N}(f(\mathbf{x}_{t,k}; \boldsymbol{\theta}_{t-1}), \nu^2 \sigma_{t,k}^2)$ if given filtration $\mathcal{F}_t$, Lemma B.1 implies that, conditioned on $\mathcal{F}_t$ and given $t, k$,

$$\Pr \left( |\widetilde{r}_{t,k} - f(\mathbf{x}_{t,k}; \boldsymbol{\theta}_{t-1})| \leq c_t \nu \sigma_{t,k} \big| \mathcal{F}_t \right) \geq 1 - \exp(-c_t^2/2).$$

Taking a union bound over $K$ arms, we have that for any $t$

$$\Pr \left( \forall k, |\widetilde{r}_{t,k} - f(\mathbf{x}_{t,k}; \boldsymbol{\theta}_{t-1})| \leq c_t \nu \sigma_{t,k} \big| \mathcal{F}_t \right) \geq 1 - K \exp(-c_t^2/2).$$

Finally, choose $c_t = \sqrt{4 \log t + 2 \log K}$ as defined in (4.1), we get the bound that

$$\Pr\left(\mathcal{E}_t^\sigma | \mathcal{F}_t\right) = \Pr\left(\forall k, |\widetilde{r}_{t,k} - f(\mathbf{x}_{t,k}; \boldsymbol{\theta}_{t-1})| \le c_t \nu \sigma_{t,k} | \mathcal{F}_t\right) \ge 1 - \frac{1}{t^2}.$$

$\square$

## B.2 PROOF OF LEMMA 4.3

Before going into the proof, some notation is needed about linear and kernelized models.

**Definition B.2.** Define $\bar{\mathbf{U}}_t = \lambda \mathbf{I} + \sum_{i=1}^t \mathbf{g}(\mathbf{x}_{i,a_i}; \boldsymbol{\theta}_0)\mathbf{g}(\mathbf{x}_{t,a_t}; \boldsymbol{\theta}_0)^\top / m$ and based on $\bar{\mathbf{U}}_t$, we further define $\bar{\sigma}_{t,k}^2 = \lambda \mathbf{g}^\top(\mathbf{x}_{t,k}; \boldsymbol{\theta}_0)\bar{\mathbf{U}}_{t-1}^{-1}\mathbf{g}(\mathbf{x}_{t,k}; \boldsymbol{\theta}_0)/m$. Furthermore, for convenience we define

$$\begin{aligned}
\mathbf{J}_t &= (\mathbf{g}(\mathbf{x}_{1,a_1}; \boldsymbol{\theta}_t) \quad \cdots \quad \mathbf{g}(\mathbf{x}_{t,a_t}; \boldsymbol{\theta}_t)), \\
\bar{\mathbf{J}}_t &= (\mathbf{g}(\mathbf{x}_{1,a_1}; \boldsymbol{\theta}_0) \quad \cdots \quad \mathbf{g}(\mathbf{x}_{t,a_t}; \boldsymbol{\theta}_0)) \\
\mathbf{h}_t &= (h(\mathbf{x}_{1,a_1}) \quad \cdots \quad h(\mathbf{x}_{t,a_t}))^\top, \\
\mathbf{r}_t &= (r_1 \quad \cdots \quad r_t)^\top, \\
\boldsymbol{\epsilon}_t &= (h(\mathbf{x}_{1,a_1}) - r_1 \quad \cdots \quad h(\mathbf{x}_{t,a_t}) - r_t)^\top,
\end{aligned}$$

where $\boldsymbol{\epsilon}_t$ is the reward noise. We can verify that $\mathbf{U}_t = \lambda \mathbf{I} + \mathbf{J}_t \mathbf{J}_t^\top / m$, $\bar{\mathbf{U}}_t = \lambda \mathbf{I} + \bar{\mathbf{J}}_t \bar{\mathbf{J}}_t^\top / m$. We further define $\mathbf{K}_t = \bar{\mathbf{J}}_t^\top \bar{\mathbf{J}}_t / m$.

The first lemma shows that the target function is well-approximated by the linearized neural network if the network width $m$ is large enough.

**Lemma B.3** (Lemma 5.1, Zhou et al. (2019)). There exists some constant $C > 0$ such that for any $\delta \in (0, 1)$, if

$$m \ge CT^4 K^4 L^6 \log(T^2 K^2 L/\delta)/\lambda_0^4,$$

then with probability at least $1 - \delta$ over the random initialization of $\boldsymbol{\theta}_0$, there exists a $\boldsymbol{\theta}^* \in \mathbb{R}^p$ such that

$$h(\mathbf{x}^i) = \langle \mathbf{g}(\mathbf{x}^i; \boldsymbol{\theta}_0), \boldsymbol{\theta}^* - \boldsymbol{\theta}_0 \rangle, \quad \sqrt{m}\|\boldsymbol{\theta}^* - \boldsymbol{\theta}_0\|_2 \le \sqrt{2\mathbf{h}^\top \mathbf{H}^{-1}\mathbf{h}} \le B, \tag{B.1}$$

for all $i \in [TK]$, where $B$ is defined in Theorem 3.5.

From Lemma B.3, it is easy to show that under this initialization parameter $\boldsymbol{\theta}_0$, we have that $\mathbf{h}_t = \bar{\mathbf{J}}_t^\top(\boldsymbol{\theta}^* - \boldsymbol{\theta}_0)$

The next lemma bounds the difference between the $\bar{\sigma}_{t,k}$ from the linearized model and the $\sigma_{t,k}$ actually used in the algorithm. Its proof, together with other technical lemmas', will be given in the next section.

**Lemma B.4.** Suppose the network size $m$ satisfies Condition 4.1. Set $\eta = C_1(m\lambda + mLT)^{-1}$, then with probability at least $1 - \delta$,

$$|\bar{\sigma}_{t,k} - \sigma_{t,k}| \le C_2\sqrt{\log m} t^{7/6} m^{-1/6} \lambda^{-2/3} L^{9/2},$$

where $C_1, C_2$ are two positive constants.

We next bound the difference between the outputs of the neural network and the linearized model.

**Lemma B.5.** Suppose the network width $m$ satisfies Condition 4.1.

Then, set $\eta = C_1(m\lambda + mLT)^{-1}$, with probability at least $1 - \delta$ over the random initialization of $\boldsymbol{\theta}_0$, we have

$$\begin{aligned}
\left|f(\mathbf{x}_{t,k}; \boldsymbol{\theta}_{t-1}) - \langle \mathbf{g}(x_{t,k}; \boldsymbol{\theta}_0), \bar{\mathbf{U}}_{t-1}^{-1}\bar{\mathbf{J}}_{t-1}\mathbf{r}_{t-1}/m \rangle\right| &\le C_2 t^{2/3} m^{-1/6} \lambda^{-2/3} L^3 \sqrt{\log m} \\
&\quad + C_3(1 - \eta m\lambda)^J \sqrt{tL/\lambda} \\
&\quad + C_4 m^{-1/6}\sqrt{\log m} L^4 t^{5/3} \lambda^{-5/3}(1 + \sqrt{t/\lambda}),
\end{aligned}$$

where $\{C_i\}_{i=1}^4$ are positive constants.

The next lemma, due to Chowdhury & Gopalan (2017), controls the quadratic value generated by an $R$-sub-Gaussian random vector $\boldsymbol{\epsilon}$:

**Lemma B.6** (Theorem 1, Chowdhury & Gopalan (2017)). Let $\{\epsilon_t\}_{t=1}^{\infty}$ be a real-valued stochastic process such that for some $R \geq 0$ and for all $t \geq 1$, $\epsilon_t$ is $\mathcal{F}_t$-measurable and $R$-sub-Gaussian conditioned on $\mathcal{F}_t$, Recall $\mathbf{K}_t$ defined in Definition B.2. With probability $0 < \delta < 1$ and for a given $\eta > 0$, with probability $1 - \delta$, the following holds for all $t$,

$$\boldsymbol{\epsilon}_{1:t}^{\top}((\mathbf{K}_t + \eta\mathbf{I})^{-1} + \mathbf{I})^{-1}\boldsymbol{\epsilon}_{1:t} \leq R^2 \log \det((1+\eta)\mathbf{I} + \mathbf{K}_t) + 2R^2 \log(1/\delta).$$

Finally, the following lemma shows the linearized kernel and the neural tangent kernel are closed:

**Lemma B.7.** For all $t \in [T]$, there exists a positive constants $C$ such that the following holds: if the network width $m$ satisfies

$$m \geq CT^6 L^6 K^6 \log(TKL/\delta),$$

then with probability at least $1 - \delta$,

$$\log\det(\mathbf{I} + \lambda^{-1}\mathbf{K}_t) \leq \log\det(\mathbf{I} + \lambda^{-1}\mathbf{H}) + 1.$$

We are now ready to prove Lemma 4.3.

*Proof of Lemma 4.3.* First of all, since $m$ satisfies Condition 4.1, then with the choice of $\eta$ ,the condition required in Lemmas B.3–B.7 are satisfied. Thus, taking a union bound, we have with probability at least $1 - 5\delta$, that the bounds provided by these lemmas hold. Then for any $t \in [T]$, we will first provide the difference between the target function and the linear function $\langle \mathbf{g}(x_{t,k};\boldsymbol{\theta}_0), \bar{\mathbf{U}}_{t-1}^{-1}\bar{\mathbf{J}}_{t-1}\mathbf{r}_{t-1}/m \rangle$ as:

$$\left| h(\mathbf{x}_{t,k}) - \left\langle \mathbf{g}(x_{t,k};\boldsymbol{\theta}_0), \bar{\mathbf{U}}_{t-1}^{-1}\bar{\mathbf{J}}_{t-1}\mathbf{r}_{t-1}/m \right\rangle \right|$$

$$\leq \left| h(\mathbf{x}_{t,k}) - \left\langle \mathbf{g}(x_{t,k};\boldsymbol{\theta}_0), \bar{\mathbf{U}}_{t-1}^{-1}\bar{\mathbf{J}}_{t-1}\mathbf{h}_{t-1}/m \right\rangle \right| + \left| \left\langle \mathbf{g}(x_{t,k};\boldsymbol{\theta}_0), \bar{\mathbf{U}}_{t-1}^{-1}\bar{\mathbf{J}}_{t-1}\boldsymbol{\epsilon}_{t-1}/m \right\rangle \right|$$

$$= \left| \left\langle \mathbf{g}(\mathbf{x}_{t,k};\boldsymbol{\theta}_0), \boldsymbol{\theta}^* - \boldsymbol{\theta}_0 - \bar{\mathbf{U}}_{t-1}^{-1}\bar{\mathbf{J}}_{t-1}\bar{\mathbf{J}}_{t-1}^{\top}(\boldsymbol{\theta}^* - \boldsymbol{\theta}_0)/m \right\rangle \right| + \left| \mathbf{g}(\mathbf{x}_{t,k};\boldsymbol{\theta}_0)^{\top}\bar{\mathbf{U}}_{t-1}^{-1}\bar{\mathbf{J}}_{t-1}\boldsymbol{\epsilon}_{t-1}/m \right|$$

$$= \left| \left\langle \mathbf{g}(\mathbf{x}_{t,k};\boldsymbol{\theta}_0), (\mathbf{I} - \bar{\mathbf{U}}_{t-1}^{-1}(\bar{\mathbf{U}}_{t-1} - \lambda\mathbf{I}))(\boldsymbol{\theta}^* - \boldsymbol{\theta}_0) \right\rangle \right| + \left| \mathbf{g}(\mathbf{x}_{t,k};\boldsymbol{\theta}_0)^{\top}\bar{\mathbf{U}}_{t-1}^{-1}\bar{\mathbf{J}}_{t-1}\boldsymbol{\epsilon}_{t-1}/m \right|$$

$$= \lambda \left| \mathbf{g}(\mathbf{x}_{t,k};\boldsymbol{\theta}_0)^{\top}\bar{\mathbf{U}}_{t-1}^{-1}(\boldsymbol{\theta}^* - \boldsymbol{\theta}_0) \right| + \left| \mathbf{g}(\mathbf{x}_{t,k};\boldsymbol{\theta}_0)^{\top}\bar{\mathbf{U}}_{t-1}^{-1}\bar{\mathbf{J}}_{t-1}\boldsymbol{\epsilon}_{t-1}/m \right|$$

$$\leq \lambda\sqrt{\mathbf{g}(\mathbf{x}_{t,k};\boldsymbol{\theta}_0)^{\top}\bar{\mathbf{U}}_{t-1}^{-1}\mathbf{g}(\mathbf{x}_{t,k};\boldsymbol{\theta}_0)}\sqrt{(\boldsymbol{\theta}^* - \boldsymbol{\theta}_0)^{\top}\bar{\mathbf{U}}_{t-1}^{-1}(\boldsymbol{\theta}^* - \boldsymbol{\theta}_0)}$$

$$\quad + \sqrt{\mathbf{g}(\mathbf{x}_{t,k};\boldsymbol{\theta}_0)^{\top}\bar{\mathbf{U}}_{t-1}^{-1}\mathbf{g}(\mathbf{x}_{t,k};\boldsymbol{\theta}_0)}\sqrt{\boldsymbol{\epsilon}_{t-1}^{\top}\bar{\mathbf{J}}_{t-1}^{\top}\bar{\mathbf{U}}_{t-1}^{-1}\bar{\mathbf{J}}_{t-1}\boldsymbol{\epsilon}_{t-1}}/m$$

$$\leq \sqrt{m}\|\boldsymbol{\theta}^* - \boldsymbol{\theta}_0\|_2\bar{\sigma}_{t,k} + \bar{\sigma}_{t,k}\lambda^{-1/2}\sqrt{\boldsymbol{\epsilon}_{t-1}^{\top}\bar{\mathbf{J}}_{t-1}^{\top}\bar{\mathbf{U}}_{t-1}^{-1}\bar{\mathbf{J}}_{t-1}\boldsymbol{\epsilon}_{t-1}}/m \tag{B.2}$$

where the first inequality uses triangle inequality and the fact that $\mathbf{r}_{t-1} = \mathbf{h}_{t-1} + \boldsymbol{\epsilon}_{t-1}$; the first equality is from Lemma B.3 and the second equality uses the fact that $\bar{\mathbf{J}}_{t-1}\bar{\mathbf{J}}_{t-1}^{\top} = m(\bar{\mathbf{U}}_{t-1} - \lambda\mathbf{I})$ which can be verified using Definition B.2; the second inequality is from the fact that $|\boldsymbol{\alpha}^{\top}\mathbf{A}\boldsymbol{\beta}| \leq \sqrt{\boldsymbol{\alpha}^{\top}\mathbf{A}\boldsymbol{\alpha}}\sqrt{\boldsymbol{\beta}^{\top}\mathbf{A}\boldsymbol{\beta}}$. Since $\mathbf{U}_{t-1}^{-1} \preceq \frac{1}{\lambda}\mathbf{I}$ and $\bar{\sigma}_{t,k}$ defined in Definition B.2, we obtain the last inequality.

Furthermore, by obtaining

$$\bar{\mathbf{J}}_{t-1}^{\top}\mathbf{U}_{t-1}^{-1}\bar{\mathbf{J}}_{t-1}/m = \bar{\mathbf{J}}_{t-1}^{\top}(\lambda\mathbf{I} + \bar{\mathbf{J}}_{t-1}\bar{\mathbf{J}}_{t-1}^{\top}/m)^{-1}\mathbf{J}_{t-1}$$

$$= \bar{\mathbf{J}}_{t-1}^{\top}(\lambda^{-1}\mathbf{I} - \lambda^{-2}\bar{\mathbf{J}}_{t-1}(\mathbf{I} + \lambda^{-1}\bar{\mathbf{J}}_{t-1}^{\top}\bar{\mathbf{J}}_{t-1}/m)^{-1}\bar{\mathbf{J}}_{t-1}^{\top}/m)\bar{\mathbf{J}}_{t-1}/m$$

$$= \lambda^{-1}\bar{\mathbf{J}}_{t-1}^{\top}\bar{\mathbf{J}}_{t-1}/m - \lambda^{-1}\mathbf{J}_{t-1}^{\top}\bar{\mathbf{J}}_{t-1}(\lambda\mathbf{I} + \bar{\mathbf{J}}_{t-1}^{\top}\bar{\mathbf{J}}_{t-1}/m)^{-1}\bar{\mathbf{J}}_{t-1}^{\top}\bar{\mathbf{J}}_{t-1}/m^2$$

$$= \lambda^{-1}\mathbf{K}_{t-1}(\mathbf{I} - (\lambda\mathbf{I} + \mathbf{K}_{t-1})^{-1}\mathbf{K}_{t-1}) = \mathbf{K}_{t-1}(\lambda\mathbf{I} + \mathbf{K}_{t-1})^{-1},$$

where the first equality is from the Sherman-Morrison formula, and the second equality uses Definition B.2 and the fact that $(\lambda\mathbf{I} + \mathbf{K}_{t-1})^{-1}\mathbf{K}_{t-1} = \mathbf{I} - \lambda(\lambda\mathbf{I} + \mathbf{K}_{t-1})^{-1}$ which could be verified by multiplying the LHS and RHS together, we have that

$$\sqrt{\boldsymbol{\epsilon}_{t-1}^{\top}\bar{\mathbf{J}}_{t-1}^{\top}\mathbf{U}_{t-1}^{-1}\bar{\mathbf{J}}_{t-1}\boldsymbol{\epsilon}_{t-1}}/m \leq \sqrt{\boldsymbol{\epsilon}_{t-1}^{\top}\mathbf{K}_{t-1}(\lambda\mathbf{I} + \mathbf{K}_{t-1})^{-1}\boldsymbol{\epsilon}_{t-1}}$$

$$\leq \sqrt{\boldsymbol{\epsilon}_{t-1}^{\top}(\mathbf{K}_{t-1} + (\lambda - 1)\mathbf{I})(\lambda\mathbf{I} + \mathbf{K}_{t-1})^{-1}\boldsymbol{\epsilon}_{t-1}}$$

$$= \sqrt{\boldsymbol{\epsilon}_{t-1}^{\top}(\mathbf{I} + (\mathbf{K}_{t-1} + (\lambda - 1)\mathbf{I})^{-1})^{-1}\boldsymbol{\epsilon}_{t-1}} \tag{B.3}$$

where the second inequality is because $\lambda = 1 + 1/T \geq 1$ set in Theorem 3.5.

Based on (B.2) and (B.3), by utilizing the bound on $\|\boldsymbol{\theta}^* - \boldsymbol{\theta}\|_2$ provided in Lemma B.3, as well as the bound given in Lemma B.6, and $\lambda \geq 1$, we have

$$\left| h(\mathbf{x}_{t,k}) - \left\langle \mathbf{g}(\mathbf{x}_{t,k}; \boldsymbol{\theta}_0), \bar{\mathbf{U}}_{t-1}^{-1} \bar{\mathbf{J}}_{t-1} \mathbf{r}_{t-1} m \right\rangle \right| \leq \left( B + R\sqrt{\log \det(\lambda \mathbf{I} + \mathbf{K}_{t-1}) + 2\log(1/\delta)} \right) \bar{\sigma}_{t,k},$$

since it is obvious that

$$\begin{aligned}
\log \det(\lambda \mathbf{I} + \mathbf{K}_{t-1}) &= \log \det(\mathbf{I} + \lambda^{-1} \mathbf{K}_{t-1}) + (t-1)\log \lambda \\
&\leq \log \det(\mathbf{I} + \lambda^{-1} \mathbf{K}_{t-1}) + t(\lambda - 1) \\
&\leq \log \det(\mathbf{I} + \lambda^{-1} \mathbf{H}) + 2,
\end{aligned}$$

where the first equality moves the $\lambda$ outside the $\log \det$, the first inequality is due to $\log \lambda \leq \lambda - 1$, and the second inequality is from Lemma B.7 and the fact that $\lambda = 1 + 1/T$ (as set in Theorem 3.5). Thus, we have

$$\left| h(\mathbf{x}_{t,k}) - \left\langle \mathbf{g}(x_{t,k}; \boldsymbol{\theta}_0), \bar{\mathbf{U}}_{t-1}^{-1} \bar{\mathbf{J}}_{t-1} \mathbf{r}_{t-1} m \right\rangle \right| \leq \nu \bar{\sigma}_{t,k},$$

where we set $\nu = B + R\sqrt{\log \det(\mathbf{I} + \mathbf{H}/\lambda) + 2 + 2\log(1/\delta)}$. Then, by combining this bound with Lemma B.5, we conclude that there exist positive constants $\bar{C}_1, \bar{C}_2, \bar{C}_3$ so that

$$\begin{aligned}
|f(\mathbf{x}_{t,k}; \boldsymbol{\theta}_{t-1}) - h(\mathbf{x}_{t,k})| &\leq \nu \bar{\sigma}_{t,k} + \bar{C}_1 t^{2/3} m^{-1/6} \lambda^{-2/3} L^3 \sqrt{\log m} + \bar{C}_2 (1 - \eta m \lambda)^J \sqrt{tL/\lambda} \\
&\quad + \bar{C}_3 m^{-1/6} \sqrt{\log m} L^4 t^{5/3} \lambda^{-5/3} (1 + \sqrt{t/\lambda}), \\
&\leq \nu \sigma_{t,k} + \bar{C}_1 t^{2/3} m^{-1/6} \lambda^{-2/3} L^3 \sqrt{\log m} + \bar{C}_2 (1 - \eta m \lambda)^J \sqrt{tL/\lambda} \\
&\quad + \bar{C}_3 m^{-1/6} \sqrt{\log m} L^4 t^{5/3} \lambda^{-5/3} (1 + \sqrt{t/\lambda}) \\
&\quad + \left( B + R\sqrt{\log \det(\mathbf{I} + \mathbf{H}/\lambda) + 2 + 2\log(1/\delta)} \right) (\bar{\sigma}_{t,k} - \sigma_{t,k}).
\end{aligned}$$

Finally, by utilizing the bound of $|\bar{\sigma}_{t,k} - \sigma_{t,k}|$ provided in Lemma B.4, we conclude that

$$|f(\mathbf{x}_{t,k}; \boldsymbol{\theta}_{t-1}) - h(\mathbf{x}_{t,k})| \leq \nu \sigma_{t,k} + \epsilon(m),$$

where $\epsilon(m)$ is defined by adding all of the additional terms and taking $t = T$:

$$\begin{aligned}
\epsilon(m) = {}& \bar{C}_1 T^{2/3} m^{-1/6} \lambda^{-2/3} L^3 \sqrt{\log m} + \bar{C}_2 (1 - \eta m \lambda)^J \sqrt{TL/\lambda} + \\
& + \bar{C}_3 m^{-1/6} \sqrt{\log m} L^4 T^{5/3} \lambda^{-5/3} (1 + \sqrt{T/\lambda}) \\
& + \bar{C}_4 \left( B + R\sqrt{\log \det(\mathbf{I} + \mathbf{H}/\lambda) + 2 + 2\log(1/\delta)} \right) \sqrt{\log m} T^{7/6} m^{-1/6} \lambda^{-2/3} L^{9/2},
\end{aligned}$$

where is exactly the same form defined in (4.3). By setting $\delta$ to $\delta/5$ (required by the union bound discussed at the beginning of the proof), we get the result presented in Lemma 4.3. $\qquad \square$

## B.3 PROOF OF LEMMA 4.4

Our proof requires an anti-concentration bound for Gaussian distribution, as stated below:

**Lemma B.8** (Gaussian anti-concentration). For a Gaussian random variable $X$ with mean $\mu$ and standard deviation $\sigma$, for any $\beta > 0$,

$$\Pr\left( \frac{X - \mu}{\sigma} > \beta \right) \geq \frac{\exp(-\beta^2)}{4\sqrt{\pi}\beta}.$$

*Proof of Lemma 4.4.* Since $\widetilde{r}_{t,k} \sim \mathcal{N}(f(\mathbf{x}_{t,k}; \boldsymbol{\theta}_{t-1}), \nu_t^2 \sigma_{t,k}^2)$ conditioned on $\mathcal{F}_t$, we have

$$
\begin{aligned}
&\Pr\left(\widetilde{r}_{t,k} + \epsilon(m) > h(\mathbf{x}_{t,k}) \middle| \mathcal{F}_t, \mathcal{E}_t^\mu\right) \\
&= \Pr\left(\frac{\widetilde{r}_{t,k} - f(\mathbf{x}_{t,k}; \boldsymbol{\theta}_{t-1}) + \epsilon(m)}{\nu \sigma_{t,k}} > \frac{h(x_{t,k}) - f(\mathbf{x}_{t,k}; \boldsymbol{\theta}_{t-1})}{\nu \sigma_{t,k}} \middle| \mathcal{F}_t, \mathcal{E}_t^\mu\right) \\
&\geq \Pr\left(\frac{\widetilde{r}_{t,k} - f(\mathbf{x}_{t,k}; \boldsymbol{\theta}_{t-1}) + \epsilon(m)}{\nu \sigma_{t,k}} > \frac{|h(x_{t,k}) - f(\mathbf{x}_{t,k}; \boldsymbol{\theta}_{t-1})|}{\nu \sigma_{t,k}} \middle| \mathcal{F}_t, \mathcal{E}_t^\mu\right) \\
&= \Pr\left(\frac{\widetilde{r}_{t,k} - f(\mathbf{x}_{t,k}; \boldsymbol{\theta}_{t-1})}{\nu \sigma_{t,k}} > \frac{|h(x_{t,k}) - f(\mathbf{x}_{t,k}; \boldsymbol{\theta}_{t-1})| - \epsilon(m)}{\nu \sigma_{t,k}} \middle| \mathcal{F}_t, \mathcal{E}_t^\mu\right) \\
&\geq \Pr\left(\frac{\widetilde{r}_{t,k} - f(\mathbf{x}_{t,k}; \boldsymbol{\theta}_{t-1})}{\nu \sigma_{t,k}} > 1 \middle| \mathcal{F}_t, \mathcal{E}_t^\mu\right) \geq \frac{1}{4e\sqrt{\pi}},
\end{aligned}
$$

where the first inequality is due to $|x| \geq x$, and the second inequality follows from event $\mathcal{E}_t^\mu$, i.e.,

$$
\forall k \in [K], \quad |f(\mathbf{x}_{t,k}; \boldsymbol{\theta}_{t-1}) - h(\mathbf{x}_{t,k})| \leq \nu \sigma_{t,k} + \epsilon(m).
$$

$\square$

### B.4 PROOF OF LEMMA 4.5

*Proof of Lemma 4.5.* Consider the following two events at round $t$:

$$
\begin{aligned}
\mathcal{A} &= \{\forall k \in S_t, \widetilde{r}_{t,k} < \widetilde{r}_{t,a_t^*} | \mathcal{F}_t, \mathcal{E}_t^\mu\}, \\
\mathcal{B} &= \{a_t \notin S_t | \mathcal{F}_t, \mathcal{E}_t^\mu\}.
\end{aligned}
$$

Clearly, $\mathcal{A}$ implies $\mathcal{B}$, since $a_t = \mathrm{argmax}_k \widetilde{r}_{t,k}$. Therefore,

$$
\Pr\left(a_t \notin S_t \middle| \mathcal{F}_t, \mathcal{E}_t^\mu\right) \geq \Pr\left(\forall k \in S_t, \widetilde{r}_{t,k} < \widetilde{r}_{t,a_t^*} \middle| \mathcal{F}_t, \mathcal{E}_t^\mu\right).
$$

Suppose $\mathcal{E}^\mu$ also holds, then it is easy to show that $\forall k \in [K]$,

$$
|h(\mathbf{x}_{t,k}) - \widetilde{r}_{t,k}| \leq |h(\mathbf{x}_{t,k}) - f(\mathbf{x}_{t,k}; \boldsymbol{\theta}_t)| + |f(\mathbf{x}_{t,k}; \boldsymbol{\theta}_t) - \widetilde{r}_{t,k}| \leq \epsilon(m) + (1 + \widetilde{c}_t)\nu_t \sigma_{t,k}. \quad \text{(B.4)}
$$

Hence, for all $k \in S_t$, we have that

$$
h(\mathbf{x}_{t,a_t^*}) - \widetilde{r}_{t,k} \geq h(\mathbf{x}_{t,a_t^*}) - h(\mathbf{x}_{t,k}) - |h(\mathbf{x}_{t,k}) - \widetilde{r}_{t,k}| \geq \epsilon(m),
$$

where we used the definitions of saturated arms in Definition 4.4, and of $\mathcal{E}_t^\mu$ and $\mathcal{E}_t^\sigma$ in (4.1).

Consider the following event

$$
\mathcal{C} = \{h(\mathbf{x}_{t,a_t^*}) - \epsilon(m) < \widetilde{r}_{t,a_t^*} | \mathcal{F}_t, \mathcal{E}_t^\mu\}.
$$

Since $\mathcal{E}_t^\sigma$ implies $h(\mathbf{x}_{t,a_t^*}) - \epsilon(m) \geq \widetilde{r}_{t,k}$, we have that if $\mathcal{C}, \mathcal{E}_t^\sigma$ holds, then $\mathcal{A}$ holds, i.e. $\mathcal{E}_t^\sigma \cap \mathcal{C} \subseteq \mathcal{A}$. Taking union with $\bar{\mathcal{E}}_t^\sigma$ we have that $\mathcal{C} = \bar{\mathcal{E}}_t^\sigma \cup \mathcal{E}_t^\sigma \cap \mathcal{C} \subseteq \mathcal{A} \cup \bar{\mathcal{E}}_t^\sigma$, which implies

$$
\Pr(\mathcal{A}) + \Pr(\bar{\mathcal{E}}_t^\sigma) \geq \Pr(\mathcal{C}). \quad \text{(B.5)}
$$

Then, (B.5) implies that

$$
\begin{aligned}
\Pr\left(\forall k \in S_t, \widetilde{r}_{t,k} < \widetilde{r}_{t,a_t^*} \middle| \mathcal{F}_t, \mathcal{E}_t^\mu\right) &\geq \Pr\left(\widetilde{r}_{t,a_t^*} + \epsilon(m) > h(\mathbf{x}_{t,a_t^*}) \middle| \mathcal{F}_t, \mathcal{E}_t^\mu\right) - \Pr\left(\bar{\mathcal{E}}_t^\sigma \middle| \mathcal{F}_t, \mathcal{E}_t^\mu\right) \\
&\geq \frac{1}{4e\sqrt{\pi}} - \frac{1}{t^2},
\end{aligned}
$$

where the first inequality is from $a_t^*$ is a special case of $\forall k \in [K]$, the second inequality is from Lemmas 4.2 and 4.4. $\square$

### B.5 PROOF OF LEMMA 4.6

To prove Lemma 4.6, we will need an upper bound bound on $\delta_{t,k}$.

**Lemma B.9.** *For any time $t \in [T]$, $k \in [K]$, and $\delta \in (0,1)$, if the network width $m$ satisfies Condition 4.1, we have, with probability at least $1 - \delta$, that*

$$\sigma_{t,k} \leq C\sqrt{L},$$

*where $C$ is a positive constant.*

*Proof of Lemma 4.6.* Recall that given $\mathcal{F}_t$ and $\mathcal{E}_t^\mu$, the only randomness comes from sampling $\widetilde{r}_{t,k}$ for $k \in [K]$. Let $\bar{k}_t$ be the unsaturated arm with the smallest $\sigma_{t,\cdot}$, i.e.

$$\bar{k}_t = \operatorname*{argmin}_{k \notin S_t} \sigma_{t,k},$$

then we have that

$$
\begin{aligned}
\mathbb{E}[\sigma_{t,a_t}|\mathcal{F}_t, \mathcal{E}_t^\mu] &\geq \mathbb{E}[\sigma_{t,a_t}|\mathcal{F}_t, \mathcal{E}_t^\mu, a_t \notin S_t] \Pr(a_t \notin S_t | \mathcal{F}_t, \mathcal{E}_t^\mu) \\
&\geq \sigma_{t,\bar{k}_t}\left(\frac{1}{4\mathrm{e}\sqrt{\pi}} - \frac{1}{t^2}\right),
\end{aligned}
\tag{B.6}
$$

where the first inequality ignores the case when $a_t \in S_t$, and the second inequality is from Lemma 4.5 and the definition of $\bar{k}_t$ mentioned above.

If both $\mathcal{E}_t^\sigma$ and $\mathcal{E}_t^\mu$ hold, then

$$\forall k \in [K], \ |h(\mathbf{x}_{t,k}) - \widetilde{r}_{t,k}| \leq \epsilon(m) + (1 + c_t)\nu\sigma_{t,k}, \tag{B.7}$$

as proved in equation (B.4). Thus,

$$
\begin{aligned}
h(\mathbf{x}_{t,a_t^*}) - h(\mathbf{x}_{t,a_t}) &= h(\mathbf{x}_{t,a_t^*}) - h(\mathbf{x}_{t,\bar{k}_t}) + h(\mathbf{x}_{t,\bar{k}_t}) - h(\mathbf{x}_{t,a_t}) \\
&\leq (1 + c_t)\nu\sigma_{t,\bar{k}_t} + 2\epsilon(m) + h(\mathbf{x}_{t,\bar{k}_t}) - \widetilde{r}_{t,\bar{k}_t} - h(\mathbf{x}_{t,a_t}) \\
&\quad + \widetilde{r}_{t,a_t} + \widetilde{r}_{t,\bar{k}_t} - \widetilde{r}_{t,a_t} \\
&\leq (1 + c_t)\nu(2\sigma_{t,\bar{k}_t} + \sigma_{t,a_t}) + 4\epsilon(m),
\end{aligned}
\tag{B.8}
$$

where the first inequality is from Definition 4.4 and $\bar{k}_t \notin S_t$, and the second inequality comes from equation (B.7). Since a trivial bound on $h(\mathbf{x}_{t,a_t^*}) - h(\mathbf{x}_{t,a_t})$ could be get by $h(\mathbf{x}_{t,a_t^*}) - h(\mathbf{x}_{t,a_t}) \leq |h(\mathbf{x}_{t,a_t^*})| + |h(\mathbf{x}_{t,a_t})| \leq 2$, then we have

$$
\begin{aligned}
\mathbb{E}[h(\mathbf{x}_{t,a_t^*}) - h(\mathbf{x}_{t,a_t})|\mathcal{F}_t, \mathcal{E}_t^\mu] &= \mathbb{E}[h(\mathbf{x}_{t,a_t^*}) - h(\mathbf{x}_{t,a_t})|\mathcal{F}_t, \mathcal{E}_t^\mu, \mathcal{E}_t^\sigma]\Pr(\mathcal{E}_t^\sigma) \\
&\quad + \mathbb{E}[h(\mathbf{x}_{t,a_t^*}) - h(\mathbf{x}_{t,a_t})|\mathcal{F}_t, \mathcal{E}_t^\mu, \bar{\mathcal{E}}_t^\sigma]\Pr(\bar{\mathcal{E}}_t^\sigma) \\
&\leq (1 + c_t)\nu(2\sigma_{t,\bar{k}_t} + \mathbb{E}[\sigma_{t,a_t}|\mathcal{F}_t, \mathcal{E}_t^\mu]) + 4\epsilon(m) + \frac{2}{t^2} \\
&\leq (1 + c_t)\nu\left(\frac{2\mathbb{E}[\sigma_{t,a_t}|\mathcal{F}_t, \mathcal{E}_t^\mu]}{\frac{1}{4\mathrm{e}\sqrt{\pi}} - \frac{1}{t^2}} + \mathbb{E}[\sigma_{t,a_t}|\mathcal{F}_t, \mathcal{E}_t^\mu]\right) + 4\epsilon(m) + \frac{2}{t^2} \\
&\leq 44\mathrm{e}\sqrt{\pi}(1 + c_t)\nu\mathbb{E}[\sigma_{t,a_t}|\mathcal{F}_t, \mathcal{E}_t^\mu] + 4\epsilon(m) + 2t^{-2},
\end{aligned}
$$

where the inequality on the second line uses the bound provide in (B.8) and the trivial bound of $h(\mathbf{x}_{t,a_t^*}) - h(\mathbf{x}_{t,a_t})$ for the second term plus Lemma 4.2, the inequality on the third line uses the bound of $\sigma_{t,\bar{k}_t}$ provide in (B.6), inequality on the forth line is directly calculated by $1 \leq 4\mathrm{e}\sqrt{\pi}$ and

$$\frac{1}{\frac{1}{4\mathrm{e}\sqrt{\pi}} - \frac{1}{t^2}} \leq 20\mathrm{e}\sqrt{\pi},$$

which trivially holds since LHS is negative when $t \leq 4$ and when $t = 5$, the LHS reach its maximum as $\approx 84.11 < 96.36 \approx \text{RHS}$.

Noticing that $|h(\mathbf{x})| \leq 1$, it is trivial to further extend the bound as

$$\mathbb{E}[h(\mathbf{x}_{t,a_t^*}) - h(\mathbf{x}_{t,a_t})|\mathcal{F}_t, \mathcal{E}_t^\mu] \leq \min\{44\mathrm{e}\sqrt{\pi}(1 + c_t)\nu\mathbb{E}[\sigma_{t,a_t}|\mathcal{F}_t, \mathcal{E}_t^\mu], 2\} + 4\epsilon(m) + 2t^{-2},$$

and since we have $1 + c_t \geq 1$ and $\nu = B + R\sqrt{\log \det(\mathbf{I} + \mathbf{H}/\lambda) + 2 + 2\log(1/\delta)} \geq B$, recall $22\mathrm{e}\sqrt{\pi}B \geq 1$, it is easy to verify the following inequality also holds:

$$
\begin{aligned}
&\mathbb{E}[h(\mathbf{x}_{t,a_t^*}) - h(\mathbf{x}_{t,a_t})|\mathcal{F}_t, \mathcal{E}_t^\mu] \\
&\leq 44\mathrm{e}\sqrt{\pi}(1 + c_t)\nu \min\{\mathbb{E}[\sigma_{t,a_t}|\mathcal{F}_t, \mathcal{E}_t^\mu], 1\} + 4\epsilon(m) + 2t^{-2} \\
&\leq 44\mathrm{e}\sqrt{\pi}(1 + c_t)\nu C_1 \sqrt{L}\mathbb{E}[\min\{\sigma_{t,a_t}, 1\}|\mathcal{F}_t, \mathcal{E}_t^\mu] + 4\epsilon(m) + 2t^{-2},
\end{aligned}
$$

where we use the fact that there exists a constant $C_1$ such that $\sigma_{t,a_t}$ is bounded by $C_1\sqrt{L}$ with probability $1 - \delta$ provided by Lemma B.9. Merging the positive constant $C_1$ with $44\mathrm{e}\sqrt{\pi}$, we get the statement in Lemma 4.6. $\qquad\square$

## B.6 Proof of Lemma 4.7

We start with introducing the Azuma-Hoeffding inequality for super-martingale:

**Lemma B.10** (Azuma-Hoeffding Inequality for Super Martingale). *If a super-martingale $Y_t$, corresponding to filtration $\mathcal{F}_t$ satisfies that $|Y_t - Y_{t-1}| \leq B_t$, then for any $\delta \in (0, 1)$, w.p. $1 - \delta$, we have*

$$
Y_t - Y_0 \leq \sqrt{2\log(1/\delta)\sum_{i=1}^{t}B_i^2}.
$$

*Proof of Lemma 4.7.* From Lemma B.9, we have that there exists a positive constant $C_1$ such that $X_t$ defined in (4.5) is bounded with probability $1 - \delta$ by

$$
\begin{aligned}
|X_t| &\leq |\bar{\Delta}_t| + C_1(1 + c_t)\nu\sqrt{L}\min\{\sigma_{t,a_t}, 1\} + 4\epsilon(m) + 2t^{-2} \\
&\leq 2 + 2t^{-2} + C_1 C_2 (1 + c_t)\nu L + 4\epsilon(m) \\
&\leq 4 + C_1 C_2 (1 + c_t)\nu L + 4\epsilon(m)
\end{aligned}
$$

where the first inequality uses the fact that $|a - b| \leq |a| + |b|$; the second inequality is from Lemma B.9 and the fact that $h \leq 1$, where $C_2$ is a positive constant used in Lemma B.9; the third inequality uses the fact that $t^{-2} \leq 1$. Noticing the fact that $c_t \leq c_T$, and from Lemma 4.6, we know that with probability at least $1 - \delta$, $Y_t$ is a super martingale. From Lemma B.10, we have

$$
Y_T - Y_0 \leq (4 + C_1 C_2 (1 + c_T)\nu L + 4\epsilon(m))\sqrt{2\log(1/\delta)T}. \tag{B.9}
$$

Considering the definition of $Y_T$ in (4.5), (B.9) is equivalent to

$$
\begin{aligned}
\sum_{i=1}^{T}\bar{\Delta}_i &\leq 4T\epsilon(m) + 2\sum_{i=1}^{T}t^{-2} + C_1(1 + c_T)\nu\sqrt{L}\sum_{i=1}^{T}\min\{\sigma_{t,a_t}, 1\} \\
&\quad + (4 + C_1 C_2 (1 + c_T)\nu L + 4\epsilon(m))\sqrt{2\log(1/\delta)T},
\end{aligned}
$$

then by utilizing $\sum_{i=1}^{\infty}t^{-2} = \pi^2/6$, and merge the constant $C_1$ with $44\mathrm{e}\sqrt{\pi}$, taking union bound of the probability bound of Lemma 4.6, B.10, B.9, we have the inequality above hold with probability at least $1 - 3\delta$. Re-scaling $\delta$ to $\delta/3$ and merging the product of $C_1 C_2$ as a new positive constant leads to the desired result. $\qquad\square$

## B.7 Proof of Lemma 4.8

We first state a technical lemma that will be useful:

**Lemma B.11** (Lemma 11, Abbasi-Yadkori et al. (2011)). *Let $\{\mathbf{v}_t\}_{t=1}^{\infty}$ be a sequence in $\mathbb{R}^d$, and define $\mathbf{V}_t = \lambda\mathbf{I} + \sum_{i=1}^{t}\mathbf{v}_i\mathbf{v}_i^\top$. If $\lambda \geq 1$, then*

$$
\sum_{i=1}^{T}\min\{\mathbf{v}_t^\top\mathbf{V}_{t-1}^{-1}\mathbf{v}_{t-1}, 1\} \leq 2\log\det\left(\mathbf{I} + \lambda^{-1}\sum_{i=1}^{t}\mathbf{v}_i\mathbf{v}_i^\top\right).
$$

*Proof of Lemma 4.8.* First, recall $\bar{\sigma}_{t,k}$ defined in Definition B.2 and the bound of $\bar{\sigma}_{t,k} - \sigma_{t,k}$ provided in Lemma B.4. We have that there exists a positive constants $C_1$ such that

$$
\sum_{i=1}^{T} \min\{\sigma_{t,a_t}, 1\} = \sum_{i=1}^{T} \min\{\bar{\sigma}_{t,a_t}, 1\} + \sum_{i=1}^{T} (\sigma_{t,a_t} - \bar{\sigma}_{t,a_t})
$$

$$
\leq \sqrt{T \sum_{i=1}^{T} \min\{\bar{\sigma}_{t,a_t}^2, 1\}} + C_1 T^{13/6} \sqrt{\log m} \, m^{-1/6} \lambda^{-2/3} L^{9/2},
$$

where the first term in the inequality on the second line is from Cauchy-Schwartz inequality, and the second term is from Lemma B.4.

From Definition B.2, we have

$$
\sum_{i=1}^{T} \min\{\bar{\sigma}_{t,a_t}^2, 1\} \leq \lambda \sum_{i=1}^{T} \min\{\mathbf{g}(\mathbf{x}_{t,a_t}, \boldsymbol{\theta}_0)^\top \bar{\mathbf{U}}_{t-1}^{-1} \mathbf{g}(\mathbf{x}_{t,a_t}, \boldsymbol{\theta}_0)/m, 1\}
$$

$$
\leq 2\lambda \log \det \left( \mathbf{I} + \lambda^{-1} \sum_{i=1}^{T} \mathbf{g}(\mathbf{x}_{t,a_t}; \boldsymbol{\theta}_0) \mathbf{g}(\mathbf{x}_{t,a_t}; \boldsymbol{\theta}_0)^\top / m \right)
$$

$$
= 2\lambda \log \det(\mathbf{I} + \lambda^{-1} \bar{\mathbf{J}}_T \bar{\mathbf{J}}_T^\top / m)
$$

$$
= 2\lambda \log \det(\mathbf{I} + \lambda^{-1} \bar{\mathbf{J}}_T^\top \mathbf{J}_T / m)
$$

$$
= 2\lambda \log \det(\mathbf{I} + \lambda^{-1} \mathbf{K}_T)
$$

where the first inequality moves the positive parameter $\lambda$ outside the $\min$ operator and uses the definition of $\bar{\sigma}_{t,k}$ in Definition B.2, then the second inequality utilizes Lemma B.11, the first equality use the definition of $\bar{\mathbf{J}}_t$ in Definition B.2, the second equality is from the fact that $\det(\mathbf{I} + \mathbf{A}\mathbf{A}^\top) = \det(\mathbf{I} + \mathbf{A}^\top \mathbf{A})$, and the last equality uses the definition of $\mathbf{K}_t$ in Definition B.2. From Lemma B.7, we have that

$$
\log \det(\mathbf{I} + \lambda^{-1} \mathbf{K}_T) \leq \log \det(\mathbf{I} + \lambda^{-1} \mathbf{H}) + 1
$$

under condition on $m$ and $\eta$ presented in Theorem 3.5. By taking a union bound we have, with probability $1 - 2\delta$, that

$$
\sum_{i=1}^{T} \min\{\bar{\sigma}_{t,a_t}, 1\} \leq \sqrt{2\lambda T (\tilde{d} \log(1 + TK) + 1)} + C_1 T^{13/6} \sqrt{\log m} \, m^{-1/6} \lambda^{-2/3} L^{9/2},
$$

where we use the definition of $\tilde{d}$ in Definition 3.2. Replacing $\delta$ with $\delta/2$ completes the proof. $\square$

## C  PROOF OF AUXILIARY LEMMAS IN APPENDIX B

In this section, we are about to show the proof of the Lemmas used in Appendix B, we will start with the following NTK Lemmas. Among them, the first is to control the difference between the parameter learned via Gradient Descent and the theoretical optimal solution to linearized network.

**Lemma C.1** (Lemma B.2, Zhou et al. (2019)). There exist constants $\{C_i\}_{i=1}^{5} > 0$ such that for any $\delta > 0$, if $\eta, m$ satisfy that for all $t \in [T]$,

$$
2\sqrt{t/\lambda} \geq C_1 m^{-1} L^{-3/2} [\log(TKL^2/\delta)]^{3/2},
$$

$$
2\sqrt{t/\lambda} \leq C_2 \min\left\{ m^{1/2} L^{-6} [\log m]^{-3/2}, m^{7/8} \left( (\lambda\eta)^2 L^{-6} t^{-1} (\log m)^{-1} \right)^{3/8} \right\},
$$

$$
\eta \leq C_3 (m\lambda + tmL)^{-1},
$$

$$
m^{1/6} \geq C_4 \sqrt{\log m} L^{7/2} t^{7/6} \lambda^{-7/6} (1 + \sqrt{t/\lambda}),
$$

then with probability at least $1 - \delta$ over the random initialization of $\boldsymbol{\theta}_0$, for any $t \in [T]$, we have that $\|\boldsymbol{\theta}_{t-1} - \boldsymbol{\theta}_0\|_2 \leq 2\sqrt{t/(m\lambda)}$ and

$$
\|\boldsymbol{\theta}_{t-1} - \boldsymbol{\theta}_0 - \bar{\mathbf{U}}_{t-1}^{-1} \bar{\mathbf{J}}_{t-1} \mathbf{r}_{t-1}/m\|_2
$$

$$
\leq (1 - \eta m\lambda)^J \sqrt{t/(m\lambda)} + C_5 m^{-2/3} \sqrt{\log m} L^{7/2} t^{5/3} \lambda^{-5/3} (1 + \sqrt{t/\lambda}).
$$

And the next lemma, controls the difference between the function value of neural network and the linearized model:

**Lemma C.2** (Lemma 4.1, Cao & Gu (2019)). There exist constants $\{C_i\}_{i=1}^3 > 0$ such that for any $\delta > 0$, if $\tau$ satisfies that

$$C_1 m^{-3/2} L^{-3/2} [\log(TKL^2/\delta)]^{3/2} \leq \tau \leq C_2 L^{-6} [\log m]^{-3/2},$$

then with probability at least $1 - \delta$ over the random initialization of $\boldsymbol{\theta}_0$, for all $\widetilde{\boldsymbol{\theta}}, \widehat{\boldsymbol{\theta}}$ satisfying $\|\widetilde{\boldsymbol{\theta}} - \boldsymbol{\theta}_0\|_2 \leq \tau, \|\widehat{\boldsymbol{\theta}} - \boldsymbol{\theta}_0\|_2 \leq \tau$ and $j \in [TK]$ we have

$$\left| f(\mathbf{x}^j; \widetilde{\boldsymbol{\theta}}) - f(\mathbf{x}^j; \widehat{\boldsymbol{\theta}}) - \langle \mathbf{g}(\mathbf{x}^j; \widehat{\boldsymbol{\theta}}), \widetilde{\boldsymbol{\theta}} - \widehat{\boldsymbol{\theta}} \rangle \right| \leq C_3 \tau^{4/3} L^3 \sqrt{m \log m}.$$

Furthermore, to continue with, next lemma is proposed to control the difference between the gradient and the gradient on the initial point.

**Lemma C.3** (Theorem 5, Allen-Zhu et al. (2018)). There exist constants $\{C_i\}_{i=1}^3 > 0$ such that for any $\delta \in (0, 1)$, if $\tau$ satisfies that

$$C_1 m^{-3/2} L^{-3/2} [\log(TKL^2/\delta)]^{3/2} \leq \tau \leq C_2 L^{-6} [\log m]^{-3/2},$$

then with probability at least $1 - \delta$ over the random initialization of $\boldsymbol{\theta}_0$, for all $\|\boldsymbol{\theta} - \boldsymbol{\theta}_0\|_2 \leq \tau$ and $j \in [TK]$ we have

$$\|\mathbf{g}(\mathbf{x}^j; \boldsymbol{\theta}) - \mathbf{g}(\mathbf{x}^j; \boldsymbol{\theta}_0)\|_2 \leq C_3 \sqrt{\log m} \tau^{1/3} L^3 \|\mathbf{g}(\mathbf{x}^j; \boldsymbol{\theta}_0)\|_2.$$

Also, we need the next lemma to control the gradient norm of the neural network with the help of NTK.

**Lemma C.4** (Lemma B.3, Cao & Gu (2019)). There exist constants $\{C_i\}_{i=1}^3 > 0$ such that for any $\delta > 0$, if $\tau$ satisfies that

$$C_1 m^{-3/2} L^{-3/2} [\log(TKL^2/\delta)]^{3/2} \leq \tau \leq C_2 L^{-6} [\log m]^{-3/2},$$

then with probability at least $1 - \delta$ over the random initialization of $\boldsymbol{\theta}_0$, for any $\|\boldsymbol{\theta} - \boldsymbol{\theta}_0\|_2 \leq \tau$ and $j \in [TK]$ we have $\|\mathbf{g}(\mathbf{x}^j; \boldsymbol{\theta})\|_F \leq C_3 \sqrt{mL}$.

Finally, as literally shows, we can also provide bounds on the kernel provided by the linearized model and the NTK kernel if the network is width enough.

**Lemma C.5** (Lemma B.1, Zhou et al. (2019)). Set $\mathbf{K} = \sum_{t=1}^T \sum_{k=1}^K \mathbf{g}(\mathbf{x}_{t,k}; \boldsymbol{\theta}_0) \mathbf{g}(\mathbf{x}_{t,k}; \boldsymbol{\theta}_0)/m$, recall the definition of $\mathbf{H}$ in Definition 3.1, then there exists a constant $C_1$ such that

$$m \geq C_1 L^6 \log(TKL/\delta) \epsilon^{-4},$$

we could get that $\|\mathbf{K} - \mathbf{H}\|_F \leq TK\epsilon$.

Equipped with these lemmas, we could continue for our proof.

## C.1 PROOF OF LEMMA B.4

*Proof of Lemma B.4.* Firstly, set $\tau = 2\sqrt{t/(m\lambda)}$, then we have the condition on the network $m$ and learning rate $\eta$ satisfy all of the condition need from Lemma C.1 to Lemma C.5. Thus from Lemma C.1, we have that there exists $\|\boldsymbol{\theta}_{t-1} - \boldsymbol{\theta}_0\|_2 \leq \tau$, thus from Lemma C.4, we have that there exists positive constant $\bar{C}_1$ such that $\|\mathbf{g}(\mathbf{x}; \boldsymbol{\theta}_{t-1})\|_2 \leq \bar{C}_1 \sqrt{mL}$, $\|\mathbf{g}(\mathbf{x}; \boldsymbol{\theta}_0)\|_2 \leq \bar{C}_1 \sqrt{mL}$, consider the function defined as

$$\psi(\mathbf{a}, \mathbf{a}_1, \cdots, \mathbf{a}_{t-1}) = \sqrt{\mathbf{a}^\top \left( \sum_{i=1}^{t-1} \lambda \mathbf{I} + \mathbf{a}_i \mathbf{a}_i^\top \right)^{-1} \mathbf{a}},$$

it is then easy to verify that

$$\psi\left( \frac{\mathbf{g}(\mathbf{x}_{t,k}; \boldsymbol{\theta}_{t-1})}{\sqrt{m}}, \frac{\mathbf{g}(\mathbf{x}_{1,a_1}; \boldsymbol{\theta}_1)}{\sqrt{m}}, \cdots, \frac{\mathbf{g}(\mathbf{x}_{t-1,a_{t-1}}; \boldsymbol{\theta}_{t-1})}{\sqrt{m}} \right) = \sigma_{t,k}$$

$$\psi\left( \frac{\mathbf{g}(\mathbf{x}_{t,k}; \boldsymbol{\theta}_0)}{\sqrt{m}}, \frac{\mathbf{g}(\mathbf{x}_{1,a_1}; \boldsymbol{\theta}_0)}{\sqrt{m}}, \cdots, \frac{\mathbf{g}(\mathbf{x}_{t-1,a_{t-1}}; \boldsymbol{\theta}_0)}{\sqrt{m}} \right) = \bar{\sigma}_{t,k},$$

then we obtain that the function $\psi$ is defined under the domain $\|\mathbf{a}\|_2 \leq \bar{C}_1\sqrt{L}, \|\mathbf{a}_i\|_2 \leq \bar{C}_1\sqrt{L}$ then by taking the derivation w.r.t. $\psi^2$, we have that

$$2\psi\partial\psi = (\partial\mathbf{a})^\top\left(\sum_{i=1}^{t-1}\lambda\mathbf{I} + \mathbf{a}_i\mathbf{a}_i^\top\right)^{-1}\mathbf{a} + \mathbf{a}^\top\left(\sum_{i=1}^{t-1}\lambda\mathbf{I} + \mathbf{a}_i\mathbf{a}_i^\top\right)^{-1}\partial\mathbf{a}$$

$$+ \mathbf{a}^\top\left(\sum_{i=1}^{t-1}\lambda\mathbf{I} + \mathbf{a}_i\mathbf{a}_i^\top\right)^{-1}\sum_{i=1}^{t-1}\left((\partial\mathbf{a}_i)\mathbf{a}_i^\top + \mathbf{a}_i\partial\mathbf{a}_i^\top\right)\left(\sum_{i=1}^{t-1}\lambda\mathbf{I} + \mathbf{a}_i\mathbf{a}_i^\top\right)^{-1}\mathbf{a},$$

by taking trace with both side and utilizing $\text{tr}(\mathbf{A}\mathbf{B}) = \text{tr}(\mathbf{B}\mathbf{A})$ and $\text{tr}(\boldsymbol{\alpha}^\top\boldsymbol{\beta}) = \text{tr}(\boldsymbol{\alpha}\boldsymbol{\beta}^\top)$, we have that

$$2\,\text{tr}(\psi\partial\psi) = \text{tr}\left(2(\partial\mathbf{a})^\top\left(\sum_{i=1}^{t-1}\lambda\mathbf{I} + \mathbf{a}_i\mathbf{a}_i^\top\right)^{-1}\mathbf{a}\right.$$

$$\left.+ 2\sum_{j=1}^{t-1}(\partial\mathbf{a}_j)^\top\left(\left(\sum_{i=1}^{t-1}\lambda\mathbf{I} + \mathbf{a}_i\mathbf{a}_i^\top\right)^{-1}\mathbf{a}\mathbf{a}^\top\left(\sum_{i=1}^{t-1}\lambda\mathbf{I} + \mathbf{a}_i\mathbf{a}_i^\top\right)^{-1}\mathbf{a}_j\right)\right),$$

thus by setting $\mathbf{C} = \left(\sum_{i=1}^{t-1}\lambda\mathbf{I} + \mathbf{a}_i\mathbf{a}_i^\top\right)^{-1}$ for simplicity and decompose $\mathbf{C} = \mathbf{Q}^\top\mathbf{D}\mathbf{Q}, \mathbf{b} = \mathbf{Q}\mathbf{a}$ where $\mathbf{D} = \text{diag}(\varrho_1, \cdots, \varrho_p)$ as the eigen-value of $\mathbf{C}$, we have that

$$\nabla_\mathbf{a}\psi = \frac{\mathbf{C}\mathbf{a}}{\sqrt{\mathbf{a}^\top\mathbf{C}\mathbf{a}}}, \|\nabla_\mathbf{a}\psi\|_2 = \sqrt{\frac{\mathbf{a}^\top\mathbf{C}^2\mathbf{a}}{\mathbf{a}^\top\mathbf{C}\mathbf{a}}} = \sqrt{\frac{\mathbf{b}^\top\mathbf{D}^2\mathbf{b}}{\mathbf{b}^\top\mathbf{D}\mathbf{b}}} = \sqrt{\frac{\sum_{i=1}^d b_i^2\varrho_i^2}{\sum_{i=1}^d b_i^2\varrho_i}} \leq 1/\sqrt{\lambda}$$

where the last inequality is from the fact that $\mathbf{C} \preceq 1/\lambda\mathbf{I}$, which indicates that all eigen-value $\varrho_i \leq 1/\lambda$, for the same reason, we have

$$\|\nabla_{\mathbf{a}_i}\psi\|_2 = \frac{\|\mathbf{C}\mathbf{a}\mathbf{a}^\top\mathbf{C}\mathbf{a}_i\|_2}{\sqrt{\mathbf{a}^\top\mathbf{C}\mathbf{a}}} \leq \|\mathbf{a}_i\|_2\frac{\|\mathbf{C}\mathbf{a}\mathbf{a}^\top\mathbf{C}\|_2}{\sqrt{\mathbf{a}^\top\mathbf{C}\mathbf{a}}} = \|\mathbf{a}_i\|_2\|\mathbf{C}\mathbf{a}\|_2\frac{\|\mathbf{C}^\top\mathbf{a}\|_2}{\sqrt{\mathbf{a}^\top\mathbf{C}\mathbf{a}}} \leq \|\mathbf{a}_i\|\|\mathbf{a}\|/\sqrt{\lambda}.$$

Thus under the domain that $\|\mathbf{a}\|_2 \leq \bar{C}_1\sqrt{L}, \|\mathbf{a}_i\|_2 \leq \bar{C}_1\sqrt{L}$, we have that

$$\|\nabla_\mathbf{a}\psi\|_2 \leq 1/\sqrt{\lambda}, \|\nabla_{\mathbf{a}_i}\psi\|_2 \leq \bar{C}_1^2 L/\sqrt{\lambda}.$$

Then, Lipschitz continuity implies

$$|\sigma_{t,k} - \bar{\sigma}_{t,k}| = \left|\psi\left(\frac{\mathbf{g}(\mathbf{x}_{t,k}; \boldsymbol{\theta}_{t-1})}{\sqrt{m}}, \frac{\mathbf{g}(\mathbf{x}_{1,a_1}; \boldsymbol{\theta}_1)}{\sqrt{m}}, \cdots, \frac{\mathbf{g}(\mathbf{x}_{t-1,a_{t-1}}; \boldsymbol{\theta}_{t-1})}{\sqrt{m}}\right)\right.$$

$$\left.- \psi\left(\frac{\mathbf{g}(\mathbf{x}_{t,k}; \boldsymbol{\theta}_0)}{\sqrt{m}}, \frac{\mathbf{g}(\mathbf{x}_{1,a_1}; \boldsymbol{\theta}_0)}{\sqrt{m}}, \cdots, \frac{\mathbf{g}(\mathbf{x}_{t-1,a_{t-1}}; \boldsymbol{\theta}_0)}{\sqrt{m}}\right)\right|$$

$$\leq \sup\{\|\nabla_\mathbf{a}\psi\|_2\}\left\|\frac{\mathbf{g}(\mathbf{x}_{t,k}; \boldsymbol{\theta}_{t-1})}{\sqrt{m}} - \frac{\mathbf{g}(\mathbf{x}_{t,k}; \boldsymbol{\theta}_0)}{\sqrt{m}}\right\|_2$$

$$+ \sum_{i=1}^{t-1}\sup\{\|\nabla_{\mathbf{a}_i}\psi\|_2\}\left\|\frac{\mathbf{g}(\mathbf{x}_{i,a_i}; \boldsymbol{\theta}_i)}{\sqrt{m}} - \frac{\mathbf{g}(\mathbf{x}_{i,a_i}; \boldsymbol{\theta}_0)}{\sqrt{m}}\right\|_2$$

$$\leq \frac{1}{\sqrt{\lambda}}\left\|\frac{\mathbf{g}(\mathbf{x}_{t,k}; \boldsymbol{\theta}_t) - \mathbf{g}(\mathbf{x}_{t,k}; \boldsymbol{\theta}_0)}{\sqrt{m}}\right\|_2 + \frac{\bar{C}_1^2 L}{\sqrt{\lambda}}\sum_{i=1}^{t-1}\left\|\frac{\mathbf{g}(\mathbf{x}_{i,a_i}; \boldsymbol{\theta}_i) - \mathbf{g}(\mathbf{x}_{i,a_i}; \boldsymbol{\theta}_0)}{\sqrt{m}}\right\|_2.$$
$$(\text{C.1})$$

By Lemma C.3 with $\tau = 2\sqrt{t/m\lambda}$, there exist positive constants $\bar{C}_2$ and $\bar{C}_3$ so that each gradient difference in (C.1) is bounded by

$$\frac{1}{\sqrt{m}}\|\mathbf{g}(\mathbf{x}; \boldsymbol{\theta}) - \mathbf{g}(\mathbf{x}; \boldsymbol{\theta}_0)\|_2 \leq \bar{C}_2\sqrt{\log m}\tau^{1/3}L^3\|\mathbf{g}(\mathbf{x}; \boldsymbol{\theta}_0)\|_2/\sqrt{m}$$

$$\leq \bar{C}_3\sqrt{\log m}t^{1/6}m^{-1/6}\lambda^{-1/6}L^{7/2}.$$

Thus, since we obtain that there exists constant $C_5$ such that

$$|\sigma_{t,k} - \bar{\sigma}_{t,k}| \le C_1 \sqrt{\log m} t^{7/6} m^{-1/6} \lambda^{-2/3} L^{9/2},$$

where we use the fact that $C_1 = \max\{\bar{C}_3, \bar{C}_3\bar{C}_1^2\}$ and $L \ge 1$ to merge the first term into the summation. This inequality is based on Lemma C.1, Lemma C.3 and Lemma C.4, thus it holds with probability at least $1 - 3\delta$. Replacing $\delta$ with $\delta/3$ completes the proof. $\quad\square$

## C.2 PROOF OF LEMMA B.5

*Proof of Lemma B.5.* Setting $\tau = 2\sqrt{t/m\lambda}$, we have the condition on the network $m$ and learning rate $\eta$ satisfy all of the condition needed by Lemmas C.1 to C.5. From Lemma C.1 we have $\|\boldsymbol{\theta}_{t-1} - \boldsymbol{\theta}_0\|_2 \le \tau$. Then, by Lemma C.2, there exists a constant $C_1$ such that

$$|f(\mathbf{x}_{t,k}; \boldsymbol{\theta}_{t-1}) - \langle \mathbf{g}(\mathbf{x}_{t,k}; \boldsymbol{\theta}_0), \boldsymbol{\theta}_{t-1} - \boldsymbol{\theta}_0 \rangle| \le C_1 t^{2/3} m^{-1/6} \lambda^{-2/3} L^3 \sqrt{\log m}, \qquad \text{(C.2)}$$

Using the bound on $\boldsymbol{\theta}_{t-1} - \boldsymbol{\theta}_0 - \bar{\mathbf{U}}_{t-1}^{-1}\bar{\mathbf{J}}_{t-1}\mathbf{r}_{t-1}/m$ provided in Lemma C.1 and the norm of gradient bound given in Lemma C.4, we have that there exist positive constants $\bar{C}_1, \bar{C}_2$ such that

$$
\begin{aligned}
&|\langle \mathbf{g}(\mathbf{x}_{t,k}; \boldsymbol{\theta}_0), \boldsymbol{\theta}_{t-1} - \boldsymbol{\theta}_0 \rangle - \langle \mathbf{g}(\mathbf{x}_{t,k}; \boldsymbol{\theta}_0), \bar{\mathbf{U}}_{t-1}^{-1}\bar{\mathbf{J}}_{t-1}\mathbf{r}_{t-1}/m \rangle| \\
&\le \|\mathbf{g}(\mathbf{x}_{t,k})\|_2 \|\boldsymbol{\theta}_{t-1} - \boldsymbol{\theta}_0 - \bar{\mathbf{U}}_{t-1}^{-1}\bar{\mathbf{J}}_{t-1}\mathbf{r}_{t-1}/m\|_2 \\
&\le \bar{C}_1 \sqrt{mL}\Big((1 - \eta m\lambda)^J \sqrt{t/(m\lambda)} + \bar{C}_2 m^{-2/3}\sqrt{\log m} L^{7/2} t^{5/3} \lambda^{-5/3}(1 + \sqrt{t/\lambda})\Big) \\
&= C_2(1 - \eta m\lambda)^J \sqrt{tL/\lambda} + C_3 m^{-1/6}\sqrt{\log m} L^4 t^{5/3} \lambda^{-5/3}(1 + \sqrt{t/\lambda}), \qquad \text{(C.3)}
\end{aligned}
$$

where $C_2 = \bar{C}_1, C_3 = \bar{C}_1\bar{C}_2$. Combining (C.2) and (C.3), we have

$$
\begin{aligned}
\left|f(\mathbf{x}_{t,k}; \boldsymbol{\theta}_{t-1}) - \langle \mathbf{g}(x_{t,k}; \boldsymbol{\theta}_0), \bar{\mathbf{U}}_{t-1}^{-1}\bar{\mathbf{J}}_{t-1}\mathbf{r}_{t-1}/m \rangle\right| &\le C_1 t^{2/3} m^{-1/6} \lambda^{-2/3} L^3 \sqrt{\log m} \\
&+ C_2(1 - \eta m\lambda)^J \sqrt{tL/\lambda} \\
&+ C_3 m^{-1/6}\sqrt{\log m} L^4 t^{5/3} \lambda^{-5/3}(1 + \sqrt{t/\lambda}),
\end{aligned}
$$

which holds with probability $1 - 3\delta$ with a union bound (Lemma C.4, Lemma C.1, and Lemma C.2). Replacing $\delta$ with $\delta/3$ completes the proof. $\quad\square$

## C.3 PROOF OF LEMMA B.7

*Proof of Lemma B.7.* From the definition of $\mathbf{K}_t$, we have that

$$
\begin{aligned}
\log\det(\mathbf{I} + \lambda^{-1}\mathbf{K}_t) &= \log\det\left(\mathbf{I} + \sum_{i=1}^{t} \mathbf{g}(\mathbf{x}_{i,a_i}; \boldsymbol{\theta}_0)\mathbf{g}(\mathbf{x}_{i,a_i}; \boldsymbol{\theta}_0)^\top/(m\lambda)\right) \\
&\le \log\det\left(\mathbf{I} + \sum_{t=1}^{T}\sum_{k=1}^{K} \mathbf{g}(\mathbf{x}_{i,a_i}; \boldsymbol{\theta}_0)\mathbf{g}(\mathbf{x}_{i,a_i}; \boldsymbol{\theta}_0)^\top/(m\lambda))\right) \\
&= \log\det(\mathbf{I} + \mathbf{K}/\lambda) \\
&\le \log\det(\mathbf{I} + \mathbf{H}/\lambda + (\mathbf{H} - \mathbf{K})\lambda) + T(\lambda - 1) \\
&\le \log\det(\mathbf{I} + \mathbf{H}/\lambda) + \langle(\mathbf{I} + \mathbf{H}/\lambda)^{-1}, (\mathbf{K} - \mathbf{H})/\lambda\rangle \\
&\le \log\det(\mathbf{I} + \mathbf{H}/\lambda) + \|(\mathbf{I} + \mathbf{H}/\lambda)^{-1}\|_F \|(\mathbf{K} - \mathbf{H})\|_F/\lambda \\
&\le \log\det(\mathbf{I} + \mathbf{H}/\lambda) + \sqrt{TK}\|(\mathbf{K} - \mathbf{H})\|_F \\
&\le \log\det(\mathbf{I} + \mathbf{H}/\lambda) + 1
\end{aligned}
$$

where the the first inequality is because the double summation on the second line contains more elements than the summation on the first line. The second inequality utilizes the definition of $\mathbf{K}$ in Lemma C.5 and $\mathbf{H}$ in Definition 3.1, the third inequality is from the convexity of $\log\det(\cdot)$ function, and the forth inequality is from the fact that $\langle \mathbf{A}, \mathbf{B} \rangle \le \|\mathbf{A}\|_F\|\mathbf{B}\|_F$. Then the fifth inequality is from the fact that $\|\mathbf{A}\|_F \le \sqrt{TK}\|\mathbf{A}\|_2$ if $\mathbf{A} \in \mathbb{R}^{TK \times TK}$ and $\lambda \ge 0$. Finally, the sixth inequality utilizes Lemma C.5 by setting $\epsilon = (TK)^{-3/2}$ with $m \ge C_1 L^6 T^6 K^6 \log(TKL/\delta)$, where we conclude our proof. $\quad\square$

## C.4    PROOF OF LEMMA B.9

*Proof of Lemma B.9.* Set $\tau$ in Lemma C.4 as $2\sqrt{t/(m\lambda)}$. Then the network width $m$ and learning rate $\eta$ satisfy all of the condition needed by Lemma C.1 to C.5. Hence, there exists $C_1$ such that $\|\mathbf{g}(\mathbf{x};\boldsymbol{\theta})\|_2 \leq \|\mathbf{g}(\mathbf{x};\boldsymbol{\theta})\|_F \leq C_1\sqrt{mL}$ for all $\mathbf{x}$, since it is easy to verify that $\mathbf{U}_t^{-1} \preceq \lambda^{-1}\mathbf{I}$. Thus we have that for all $t \in [T], k \in [K]$,

$$\sigma_{t,k}^2 = \lambda\mathbf{g}^\top(\mathbf{x}_{t,k};\boldsymbol{\theta}_{t-1})\mathbf{U}_{t-1}^{-1}\mathbf{g}(\mathbf{x}_{t,k};\boldsymbol{\theta}_{t-1})/m \leq \|\mathbf{g}(\mathbf{x}_{t,k};\boldsymbol{\theta}_{t-1})\|_2^2/m \leq C_5^2 L.$$

Therefore, we could get that $\sigma_{t,k} \leq C_1\sqrt{L}$, with probability $1 - 2\delta$ by taking a union bound (Lemmas C.1 and C.4). Replacing $\delta$ with $\delta/2$ completes the proof. $\qquad\square$

## D    AN UPPER BOUND OF EFFECTIVE DIMENSION $\widetilde{d}$

We now provide an example, showing when all contexts $\mathbf{x}_i$ concentrate on a $d'$-dimensional non-linear subspace of the RKHS space spanned by NTK, the effective dimension $\widetilde{d}$ is bounded by $d'$. We consider the case when $\lambda = 1, L = 2$. Suppose that there exists a constant $d'$ such that for any $i > d', 0 < \lambda_i(\mathbf{H}) \leq 1/(TK)$. Then the effective dimension $\widetilde{d}$ can be bounded as

$$\widetilde{d} = \frac{\log\det(\mathbf{I}+\mathbf{H})}{\log(1+TK)} \leq \sum_{i=1}^{TK}\log(1+\lambda_i(\mathbf{H})) \leq \sum_{i=1}^{TK}\lambda_i(\mathbf{H}) = \underbrace{\sum_{i=1}^{d'}\lambda_i(\mathbf{H})}_{I_1} + \underbrace{\sum_{i=d'+1}^{TK}\lambda_i(\mathbf{H})}_{I_2}.$$

For $I_1$ and $I_2$ we have

$$I_1 \leq \sum_{i=1}^{d'}\|\mathbf{H}\|_2 = \Theta(d'), \ I_2 \leq TK \cdot 1/(TK) = 1,$$

Therefore, the effective dimension satisfies that $\widetilde{d} \leq d' + 1$. To show how to satisfy the requirement, we first give a charcterization of the RKHS space spanned by NTK. By Bietti & Mairal (2019); Cao et al. (2019) we know that each entry of $\mathbf{H}$ has the following formula:

$$\mathbf{H}_{i,s} = \sum_{k=0}^{\infty}\mu_k \sum_{j=1}^{N(d,k)} Y_{k,j}(\mathbf{x}_i)Y_{k,j}(\mathbf{x}_s),$$

where $Y_{k,j}$ for $j = 1,\ldots,N(d,k)$ are linearly independent spherical harmonics of degree $k$ in $d$ variables, $d$ is the input dimension, $N(d,k) = (2k+d-2)/k \cdot C_{k+d-3}^{d-2}$, $\mu_k = \Theta(\max\{k^{-d}, (d-1)^{-k+1}\})$. In that case, the feature mapping $(\sqrt{\mu_k}Y_{k,j}(\mathbf{x}))_{k,j}$ maps any context $\mathbf{x}$ from $\mathbb{R}^d$ to a RKHS space $\mathcal{R}$ corresponding to $\mathbf{H}$. Let $\mathbf{y}_i \in \mathcal{R}$ denote the mapping for $\mathbf{x}_i$. Then if there exists a $d'$-dimension subspace $\mathcal{R}'$ such that for all $i$, $\|\mathbf{y}_i - \mathbf{z}_i\| \ll 1$ where $\mathbf{z}_i$ is the projection of $\mathbf{y}_i$ onto $\mathcal{R}_{d'}$, the requirement for $\lambda_i(\mathbf{H})$ holds.

