# OpenReview forum: "Neural Thompson Sampling"
_ICLR.cc/2021/Conference — ICLR 2021 Poster_

### Official Review · AnonReviewer1 · 2020-10-28
**Good paper overall; would have liked more discussion on the assumptions**

**Rating:** 7
**Confidence:** 3

**Review:**

The paper proposes neural thompson sampling (TS) - a method to run TS without assuming that the reward is a linear function of the context, as is generally assumed in literature. This is not the first paper to use neural networks for TS, however existing papers either a) used TS only in the last layer, or b) maintained uncertainty over the weights and sampled the entire neural network. This paper instead maintains a single network that computes the mean of the reward distribution of an arm.

The paper also is the first paper to provide regret guarantees for a neural TS algorithm. Experiments show that their algorithm performs better than other baselines.

My concern with the theoretical results is a missing discussion on their utility with respect to the assumptions. The necessary assumption for all results in the paper is Condition 4.1, which assumes that m, the width of the network is larger than T^6 L^6 K^6. With T as the horizon, this assumes a neural network width of 10^18 even for a modest horizon of 1000 (as used in the experiments). I don't think the experiments used this width in their implementation. I would like if the authors point out this disconnect for the benefit of the readers, and have a discussion section. I believe this assumption may be necessitated by the use of NTK.

Second, the algorithm uses the function g to model the variance of the distribution, but I did not find any discussion. Is g assumed to be known? If not, how is it learned?

I like that the authors study the delayed reward experiments as it is often the case in practical situations. What will also be useful is to discuss the implementation complexity (computation required to decide the next arm to be sampled) of various algorithms (ideally through a plot). Some algorithms may be faster than others, and readers can use this plot to make an informed choice.

---

> ### Author Response · Authors · 2020-11-18
> **Response to Reviewer 1**
>
> Thanks for your detailed comments and constructive suggestions.
>
> Q1: Is there a disconnect between theory and practice regarding the network width $m$?
>
> A1: We agree with the reviewer that the NTK theory requires a large value of $m$, while in practice $m$ can be much smaller (as in our experiments).  The disconnection has its root in the current deep learning theory based on the neural tangent kernel and is not specific to our work.  We have added a discussion on this point in Remark 3.8 and Section A.1.
>
>
> Q2: Definition of $\mathbf{g}$.
>
> A2: $\mathbf{g}$ is the gradient of the neural network, i.e., $\mathbf{g}(\mathbf{x}; \boldsymbol{\theta}) := \nabla_{\boldsymbol{\theta}}f(\mathbf{x}; \boldsymbol{\theta})$.  We have added the definition in the revision.
>
>
> Q3: How is the computation required by these bandit algorithms?
>
> A3: Thank you for your suggestion. We have added a run time plot (Figure 5 in Appendix A.3) to compare the implementation complexity of these algorithms. According to Figure 5, Neural TS and Neural UCB are both 2~3 times slower than the $\epsilon$-greedy algorithm which is caused by calculating the gradient of the network parameter for each input context. Moreover, BootstrapNN is about 5+ times slower than $\epsilon$-greedy, since it needs to train several networks in each round.

---

### Official Review · AnonReviewer4 · 2020-10-28
**Novel and efficient Thompson sampling algorithm for neural networks**

**Rating:** 7
**Confidence:** 4

**Review:**

Summary:

The paper proposes a novel Thompson sampling algorithm for neural networks which can be applied to any arbitrary, bounded reward function. While existing works apply Thompson sampling (TS) to neural networks in a heuristic way (e.g., sampling parameters in the last layer only), this algorithm considers the posterior distribution of all the parameters and is the first to provide a theoretical, tight regret upper bound. The work builds on the neural tangent kernel theory, which enables the use of techniques developed for linear reward functions (Agrawal and Goyal, 2013). Actually, the paper is analogous to the work of Zhou et al. (2020) which proposed the NeuralUCB algorithm by combining the neural tangent kernel theory and the Linear UCB methodology (Abbasi-Yadkori et al., 2011).

Reason for Score:

I vote for accepting. I believe the contribution of the paper is significant, as it is the first to construct a valid posterior distribution for the parameters in the neural networks. The method also achieves a tight regret upper bound that matches the bound of NeuralUCB. The experiments show that empirically, neuralTS performs better than the state of the art, neuralUCB.

Pros

The paper provides an efficient way for doing directed exploration in bandits with neural network reward models.
The presentation of the algorithm, theorem, and experiments is clear and concrete.

Cons

According to the proofs, the neural TS algorithm that uses initial parameter \theta_0 instead of \theta_t should have smaller regret than the current algorithm. That is to say, according to the proofs, the regret can be decomposed as (the regret of the algorithm which does not update \theta ) + (a term related to the difference of \theta_t and \theta_0).  I think the authors should add discussion about this point.

Minor comments

 In the last line of page 1:
“Sampling the estimated reward from posterior” is a confusing statement. Actually, we sample a parameter from the posterior. The same goes for “sampling the reward” between equations (2.2) and (2.3)

In the last line of Related work, authors mention Foster and Rhaklin. I don’t think they use UCB exploration. They also use a randomized algorithm although it isn't TS.

---

> ### Author Response · Authors · 2020-11-18
> **Response to Reviewer 4**
>
> Thanks for your positive and helpful comments.
>
> Q1: Can you address the decomposition of regret between $\boldsymbol{\theta}$ and $\boldsymbol{\theta}_0$?
>
> A1: We believe there is a misunderstanding. According to our initialization scheme described in line 2 of Algorithm 1, and the construction of the context vector described in Assumption 3.4, the output of the initial neural network for any context vector is 0. Therefore, only using the initial neural network (of parameter $\boldsymbol{\theta}_0$) without updating the parameter will lead to a linear regret.
>
>
> Q2: “Sampling the estimated reward from posterior” is a confusing statement
>
> A2: We would like to clarify that Neural TS indeed samples from the reward posterior distribution, rather than sampling the parameter, as described in Line 6 of Algorithm 1.  This is one of the differences from previous Thompson Sampling algorithms. In specific, we sample from a Gaussian distribution of reward, where the mean is the estimated reward by the neural network. Then we pull the arm by taking the argmax of the sampled value. Therefore, “Sampling the estimated reward from posterior” is a correct description of the algorithm.
>
> Q3: Foster and Rhaklin’s work is neither a UCB exploration nor a TS exploration.
>
> A3: Thank you for pointing it out. We have revised the description.

---

### Official Review · AnonReviewer3 · 2020-10-28
**Paper is marginally above the acceptace threshold**

**Rating:** 7
**Confidence:** 3

**Review:**

*****  Paper's Summary  *****

The authors proposed an algorithm named Neural Thompson Sampling (NeuralTS) for solving contextual multi-armed bandit problems. NeuralTS uses deep neural networks for dealing with exploration and exploitation. In the paper, the authors proved the sub-linear regret of NeuralTS, which is also verified using experiments.


*****  Paper's Strengths  *****

As NeuralTS uses the deep neural network, it can be used for estimating non-linear reward function in the contextual bandits problem.

The authors proved the sub-linear regret using the recent theoretical results from deep learning. The regret upper bound is similar (in terms of the number of contexts and effective dimension) to the regret bound of existing methods.

The performance of NeuralTS matches with the state-of-the-art baselines. In some cases, the performance is even better than the existing methods.


*****  Paper's Weaknesses  *****

The weak point of the paper is its novelty. The result incorporates recent deep learning and contextual bandits results [Zhou et al. 2019] (paper in ICML 2020) with the existing Thompson Sampling variant for contextual bandits problem.

Since the neural network (parameters m and L) is fixed before using the algorithm, it may not be possible to estimate any arbitrary reward function with the fixed neural network. Therefore, NeuralTS can have linear regret for the cases where the reward function can not be estimated.


*****  Comments  *****

It is difficult to understand the second part of Assumption 3.4. More clarity may help readers.

The experiments can be repeated 50 or more times to get a better confidence interval.


*****  Questions for the Authors  *****

Please address the above weaknesses of the paper.

How are the values of '\nu,' 'L,' and 'm' set in the experiments?

Why does NeuralTS need T as input?


***** Post Rebuttal *****

I thank authors for the clarifications! After reading the rebuttal and comments of other reviewers, I am increasing my score.

---

> ### Author Response · Authors · 2020-11-18
> **Response to Reviewer 3**
>
> Thanks for your detailed comments and suggestions.
>
> Q1: Technical novelty compared to Neural UCB and other TS works
>
> A1: Firstly, from linear to kernel version, TS and UCB are similar to each other. Thus it is not surprising that our result is similar to the Neural UCB one. However, our contribution is not a trivial combination of the TS analysis and NTK: if we directly apply the Linear TS analysis, then directly perturbing the neural network parameter will make the neural network parameter out of the regime of NTK, which will make it difficult to do the analysis. Furthermore, since our covariance matrix is calculated via the gradient of the network at time $t$, which makes the analysis of Kernel TS not applicable to our algorithm. To solve this problem, we introduce some novel techniques to deal with the intrinsic randomness from neural network learning and the randomness from the sampling distribution, as we have done in Lemmas 4.2 and 4.3. Therefore, we believe that our result is not a trivial combination of Neural UCB’s result with other TS tricks in the bandit literature. From the practical perspective, since TS is preferred in various application scenarios when action randomization is critical, as we have shown in the delayed reward section, we believe it is important to have a Neural TS algorithm with a provable regret guarantee.
>
> Q2: Since the neural network (parameters m and L) is fixed before using the algorithm, it may not be possible to estimate any arbitrary reward function with the fixed neural network. Therefore, NeuralTS can have linear regret for the cases where the reward function can not be estimated.
>
> A2: The reviewer is right that the $\sqrt{T}$ regret does not hold for arbitrary reward functions. As discussed in Remark 3.6, a sufficient condition to guarantee $\sqrt{T}$ regret is that $h$ lies in the function space induced by NTK.
>
>
> Q3: How are the values of '\nu,' 'L,' and 'm' set in the experiments?
>
> A3: We describe how to set these parameters in Section 5 and Appendix A of the original submission (as well as the current version): $L=2$, $m=100$, and $\nu$ is selected by a grid search on $\{1, 10^{-1}, 10^{-2}, \cdots, 10^{-5}\}$. For most environments, $\nu = 10^{-4}$ gives good performance. Furthermore, we also tried $m=1000$ but the results are similar to those of $m=100$.
>
>
> Q4: Second part of Assumption 3.4.
>
> A4: The duplication assumption made in contexts ensures the initial output of the neural network $f(\mathbf{x}; \boldsymbol{\theta})$ is $0$ upon initialization, as described by the paragraph right after Assumption 3.4.
>
>
> Q5: Experiments can be repeated 50 times.
>
> A5: Thank you for your suggestion. Since neural networks training takes a long time, just as other neural network methods, we have tried our best to repeat our experiments 20 times during the author’s response. We have updated our results in Figures 1 and 3. The relationship between the performance of these algorithms remains unchanged. We will repeat the experiments 50 times in the camera-ready of our paper.
>
> Q6: Why does Neural TS need input $T$?
>
> A6: Theorem 3.5 suggests that, in order to obtain a sublinear regret, the network width $m$ needs to satisfy that $m \geq \text{poly}(T)$. Therefore, we need to know $T$ at the beginning in order to set $m$. However, we can get rid of the dependence on $T$ by the doubling trick (Cesa-Bianchi & Lugosi, 2006). To do that, we decompose the time interval $(0, +\infty)$ as a union of several disjoint intervals $[2^s, 2^{s+1})$. When $2^s \leq t < 2^{s+1}$, we run NeuralTS with the input $T = 2^{s+1}$. It can be verified that similar $\tilde O(\tilde d\sqrt{t}))$ regret still holds.  We have added a discussion about this as Remark 3.9 in the revision.

---

### Official Review · AnonReviewer2 · 2020-10-29
**Interesting idea, incremental contribution**

**Rating:** 6
**Confidence:** 5

**Review:**

This work proposes a neural network based thompson sampling algorithm for general bounded reward contextual bandit problems.  They provide a theoretical regret guarantee for the proposed algorithm with the help of recent advances of Neural Tanget Kernel.

The main techniques that use the gradient of the neural networks and NTK have been developed in a recent paper, Neural UCB, as cited by the authors as well. Based on estimations of mean and variance developed in the Neural UCB, NeuralTS is a variant that uses a Gaussian posterior on the top of those estimations. The theoretical contribution is adapting some well-known tricks in the bandit literature to analyze the regret of TS algorithm. That is why I think the contribution is quite incremental.

The authors provide sufficient experimental data points to justify the advances of the proposed Neural TS against all the existing benchmarks, which is a plus.

Besides, I have the following detailed comments:

1. In Section 2, I didn't find a clear definition of $m$. I have to infer that it is the nework width until Algorithm 1.
2. In the entire paper, I didn't find a definition for $g(x,\theta)$. I have to infer its meaning by reading Neural UCB paper.
3. In Theorem 3.5, many parameters depend on the time horizon T. However, in some use case, we may not have the information of T and some algorithms have this advantage of so-called any-time regret bound. Is it possible to get rid of this dependence by playing some doubling trick?

---

> ### Author Response · Authors · 2020-11-18
> **Response to Reviewer 2**
>
> Thanks for your detailed comments and suggestions.
>
> Q1: Definition of $m$
>
> A1: $m$ is the neural network width. We have added the definition in the revision.
>
> Q2: Definition of $\mathbf{g}$
>
> A2: $\mathbf{g}$ is the gradient of the neural network, i.e., $\mathbf{g}(\mathbf{x}; \boldsymbol{\theta}) := \nabla_{\boldsymbol{\theta}}f(x; \boldsymbol{\theta})$.  We have added the definition in the revision.
>
> Q3: The contribution is incremental
>
> A3: We disagree respectfully.  Firstly, as in the linear and kernelized cases, in neural bandit, TS and UCB share some similarities but they rely on different tools in the regret analysis.  Second, our contribution is not a trivial combination of existing TS analysis and NTK: if we directly apply the Linear TS analysis, then directly perturbing the neural network parameter will push the neural network parameter out of the regime that can be handled by existing theoretical tools of NTK. This requires us to randomize the reward rather than the parameters.  Third, since our covariance matrix is calculated via the gradient of the network at time $t$, analysis of Kernel TS is not applicable to our algorithm because we cannot get direct access to the NTK kernel. Instead, we can only approximate the covariance matrix from $\mathbf{g}(\mathbf{x}; \boldsymbol{\theta})$. To solve this problem, we introduce some novel techniques to deal with the intrinsic randomness from neural network learning and the randomness from the sampling distribution, as we have done in Lemmas 4.2 and 4.3. Finally, from a practical perspective, TS is preferred in various application scenarios over deterministic strategies like UCB when action randomization is critical, as we have shown in the delayed reward section. Therefore, we believe it is important to have a Neural TS algorithm with a provable regret guarantee.
>
> Q4: Many parameters depend on $T$. Is it possible to get rid of the dependence $T$ by playing some doubling trick?
>
> A4: Yes, it is possible to get rid of the dependence on $T$ by the doubling trick. Recall that Theorem 3.5 suggests that to obtain a sublinear regret, the network width $m$ needs to satisfy that $m \geq \text{poly}(T)$. Therefore, we need to know $T$ before we run the algorithm to set $m$. For the case where $T$ is unknown, we can use the standard doubling trick (e.g., Cesa-Bianchi & Lugosi, 2006) to set $m$ adaptively.  To do that, we decompose the time interval $(0, +\infty)$ as a union of non-overlapping intervals $[2^s, 2^{s+1})$. When $2^s \leq t < 2^{s+1}$, we run NeuralTS with the input $T = 2^{s+1}$. It can be verified that similar $\tilde O(\tilde d\sqrt{t})$ regret still holds.  We have added a discussion about this as Remark 3.9 in the revision.

---

### Author Response · Authors · 2020-11-18
**Response to All Reviewers**

We thank all reviewers for their helpful feedback and constructive suggestions. We have addressed all your questions and concerns and updated our paper. The updates in the revised paper are highlighted in red color.

---

### Decision · Program_Chairs · 2021-01-07
**Final Decision**

**Decision:**

Accept (Poster)

**Comment:**

All reviewers tend towards accepting the paper, and I agree.